# ADVERSARIAL COLLABORATIVE LEARNING ON NON-IID FEATURES

## ABSTRACT

Federated Learning (FL) has been a popular approach to enable collaborative learning on multiple parties without exchanging raw data. However, the model performance of FL may degrade a lot due to non-IID data. While many FL algorithms focus on non-IID labels, FL on non-IID features has largely been overlooked. Different from typical FL approaches, the paper proposes a new learning concept called ADCOL (**Ad**versarial **Co**llaborative **L**earning) for non-IID features. Instead of adopting the widely used model-averaging scheme, ADCOL conducts training in an adversarial way: the server aims to train a discriminator to distinguish the representations of the parties, while the parties aim to generate a common representation distribution. Our experiments show that ADCOL achieves better performance than state-of-the-art FL algorithms on non-IID features.

## 1 INTRODUCTION

Deep learning is data hungry. While data are always dispersed in multiple parties (e.g., mobile devices, hospitals) in reality, data are not allowed to transfer to a central server for training due to privacy concerns and data regulations. Collaborative learning among multiple parties without the exchange of raw data has been an important research topic.

Federated learning (FL) (McMahan et al., 2016; Kairouz et al., 2019; Li et al., 2019b;a) has been a popular form of collaborative learning without exchanging raw data. A basic FL framework is FedAvg (McMahan et al., 2016), which uses a model-averaging scheme. In each round, the parties update their local models and send them to the server. The server averages all local models to update the global model, which is sent back to the parties as the new local model in the next round. FedAvg has been widely used due to its effectiveness and simpleness. Most existing FL approaches are designed based on FedAvg.

However, as shown in many existing studies (Hsu et al., 2019; Li et al., 2020; 2021a), the performance of FedAvg and its alike algorithms may be significantly degraded in non-IID data among parties. While many studies try to improve FedAvg on non-IID data, most of them (Li et al., 2020; Wang et al., 2020b; Karimireddy et al., 2020; Acar et al., 2021; Li et al., 2021b; Wang et al., 2020a) focus on the label imbalance setting, where the parties have different label distributions. In their experiments, they usually simulate the federated setting by unbalanced partitioning the dataset into multiple subsets according to labels.

As summarized in Hsieh et al. (2020); Kairouz et al. (2019), besides the label distribution skew, feature imbalance is also an important case of non-IID data. In the feature imbalance setting, the feature distribution $P_i(\mathbf{x})$ varies across parties. This setting widely exists in reality, e.g., people have different stroke width and slant when writing the same word. Another example in practice is that images collected by different cameras have different intensity and contrast. However, compared with non-IID labels, FL on non-IID features has been less explored. Most existing studies on non-IID data are still based on the model-averaging scheme (Li et al., 2020; Collins et al., 2021; Li et al., 2021b; Fallah et al., 2020), which implicitly assumes that the local knowledge $P_i(y|x)$ is common across parties and is not applicable in the non-IID feature setting. For example, FedRep (Collins et al., 2021) learns a common base encoder among parties, which will output very different representation distributions across parties in the non-IID feature case even though for the data from the same class. Such a model-sharing design fails to achieve good model accuracy for application scenarios

with non-IID features. Therefore, we need a fundamentally new approach to address the technical challenges of non-IID features.

In this paper, we think out of the model-averaging scheme used in FL, and propose a novel learning concept called adversarial collaborative learning. While the feature distribution of each party is different, we aim to extract the common representation distribution that is sufficient for the prediction task. Instead of averaging the local models, we apply adversarial learning to match the representation distributions of different parties. Specifically, the server aims to train a discriminator to distinguish the local representations by the party IDs, while the parties train the base encoders such that the generated representations cannot be distinguished by the discriminator. Besides the base encoders, each party trains a predictor for local personalization and ensures that the generated representation is meaningful for the prediction task. Our experiments show that ADCOL outperforms state-of-the-art FL algorithms on three real-world tasks. More importantly, ADCOL points out a promising research direction on collaborative learning. For example, it is interesting to generalize ADCOL to other settings besides feature skew in a communication-efficient way.

## 2 BACKGROUND AND RELATED WORK

### 2.1 NON-IID DATA

We use $P_i(\mathbf{x}, y)$ to denote the data distribution of party $i$, where $\mathbf{x}$ is the features and $y$ is the label. According to existing studies (Kairouz et al., 2019; Hsieh et al., 2020), we can categorize non-IID data in FL into the following four classes: (1) non-IID labels: the marginal distribution $P_i(y)$ varies across parties. (2) non-IID features: the marginal distribution $P_i(\mathbf{x})$ varies across parties. (3) concept drift: The conditional distributions $P_i(y|\mathbf{x})$ or $P_i(\mathbf{x}|y)$ varies across parties. (4) quantity skew: the amount of data varies across parties. In this paper, we focus on non-IID features, which widely exist in reality. For example, the distributions of images collected by different camera devices may vary due to the different equipment and environments.

### 2.2 FEDERATED LEARNING ON NON-IID LABELS

Non-IID data is a key challenge in FL. There have been many studies trying to improve the performance of FL under non-IID data. However, most existing approaches (Li et al., 2020; Wang et al., 2020a; Hsu et al., 2019; Li et al., 2021b; Acar et al., 2021; Karimireddy et al., 2020; Wang et al., 2021; Luo et al., 2021; Mendieta et al., 2022) simulate the federated setting with heterogeneous label distributions in the experiments, which does not pay attention to the non-IID feature challenge. For example, FedProx (Li et al., 2020) introduces a proximal term in the objective of local training, which limits the update of the local model by the distance between the local model and the global model. While it is challenging to achieve a good global model for every party, personalized FL (Fallah et al., 2020; Dinh et al., 2020; Hanzely et al., 2020; Zhang et al., 2021b; Huang et al., 2021; Collins et al., 2021) is a very promising direction, which aims to learn a personalized local model for each party. For example, FedRep (Collins et al., 2021) only adopts federated averaging for the base encoder, while each party locally trains a classifier head for personalization. Per-FedAvg (Fallah et al., 2020) applies the idea of model-agnostic meta-learning (Finn et al., 2017), which finds a shared model that can be easily adapted to the local datasets with a few steps of gradient descent. However, the above approaches are all based on the model-averaging scheme, which is not suitable for the non-IID feature setting as we will show in Section 3.2. They have severe performance degradation on parties with non-IID features.

### 2.3 FEDERATED LEARNING ON NON-IID FEATURES

Only several studies investigate FL on non-IID feature setting. Observing that averaging batch normalization parameters may decrease the accuracy a lot, FedBN (Li et al., 2021c) updates all the batch normalization (BN) parameters locally and does not synchronize them with the global model. The operations for non-BN parameters are the same as FedAvg. Considering each party as a domain, cross-domain FL (Sun et al., 2021) is also applicable in the non-IID feature setting. Besides BN parameters, PartialFed (Sun et al., 2021) updates selective model parameters locally and does not initialize them as the global model. While both studies try to address the feature skew problem by

partially averaging the local models, we propose a fundamentally new training framework based on adversarial learning that does not average the models at all.

## 2.4 ADVERSARIAL LEARNING FOR DISTRIBUTION MATCHING

Adversarial learning has been successful for distribution matching (e.g., domain adaptation (Tzeng et al., 2017), GANs (Goodfellow et al., 2014)). The basic idea is to train a discriminator to encourage indistinguishable distributions, which is smart and sweet. Peng et al. (2019b) proposes FADA to apply adversarial learning in federated setting to study the federated domain adaption problem, which has a different setting and goal from our paper. For more discussion on the relation and difference between our approach and FADA, please refer to Appendix C. More recently, a study (Zhang et al., 2021a) proposed FedUFO, where each party trains a discriminator to apply feature and objective consistency constrains to address the non-IID data issue. However, during the local training stage, FedUFO needs to transfer each local model to all the other parties, which causes massive communication overhead. Moreover, FedUFO focuses on the non-IID label setting in their experiments.

## 3 THE PROPOSED METHOD

### 3.1 PROBLEM STATEMENT

Suppose there are $N$ parties, where party $i$ has a local dataset $\mathcal{D}^i = \{\mathbf{x}, y\}$. The feature distributions $P(\mathbf{x})$ are different among parties while the label distributions $P(y)$ are same/similar among parties. The parties conduct collaborative learning over $\mathcal{D} \triangleq \bigcup_{i \in [N]} \mathcal{D}^i$ with the help of a central server without exchanging the raw data. Like typical personalized FL, the goal of each party is to train a machine learning model which has good accuracy on its local test dataset.

### 3.2 MOTIVATION

**Problem of Model-Averaging on non-IID Features**  Most existing studies (Li et al., 2020; Karimireddy et al., 2020; Li et al., 2021b; Dinh et al., 2020; Collins et al., 2021) are still based on FedAvg to address the non-IID data. However, they are not suitable in our setting. In the model-averaging scheme, the server averages the local models as a global model, which essentially tries to learn a common $P(y|\mathbf{x})$. In their experiments, they usually partition a dataset to different parties horizontally to simulate the federated setting, where parties indeed follow the same $P(y|\mathbf{x})$ (with different $P_i(y)$) so that the global model is helpful. However, in our setting, for party $i$ and $j$, since $P_i(\mathbf{x}) \neq P_j(\mathbf{x})$ and $P_i(y) = P_j(y)$, $P_i(y|\mathbf{x})$ and $P_j(y|\mathbf{x})$ are different. Averaging the local models does not directly help the learning of local knowledge $P_i(y|\mathbf{x})$.

Instead of averaging and learning a global model, we propose to learning a common representation distribution to address the non-IID features. Although the feature distributions $P_i(\mathbf{x})$ are different among parties, they have the same task $y$. Thus, we aim to extract the underlying task-specific representation $\mathbf{z}$ for the task $y$ from multiple parties. Specifically, we decompose local objective $P_i(y|\mathbf{x})$ into two parts: $P_i(\mathbf{z}|\mathbf{x})$ and $P_i(y|\mathbf{z})$. The first part is to learn the oracle representation for the task and the second part is to predict the label by the representations. The second part can be easily achieved by training a predictor head with the representations as inputs. For the first part, we assume that there exists an oracle optimal representation distribution $P^*(\mathbf{z})$ for the prediction of $y$. Then, the ideal objective of party $i$ can be formulated as

$$\min_{\theta_i} E_{\mathbf{x} \sim D_i} \ell_{KL}((P_i(\mathbf{x})P_i(\mathbf{z}|\mathbf{x}; \theta_i)) \parallel P^*(\mathbf{z})), \tag{1}$$

where $\ell_{KL}$ is the KL divergence loss and $\theta_i$ is the base encoder to generate the representation. In practice, $P^*(\mathbf{z})$ is unknown. However, it has the following two features: (1) $P^*(\mathbf{z})$ is same for each party; (2) $P^*(\mathbf{z})$ is able to predict $y$. Thus, we approximate the objective by two aspects: (1) To ensure that the representation absorbs the knowledge of multiple parties, the parties aim to map their local data into a common representation distribution $P(\mathbf{z})$; (2) We ensure that the generated $P(\mathbf{z})$ contains necessary information for the prediction of $y$ by training a predictor on the representation. We introduce the details about the training procedure in Section 3.4.

### 3.3 MODEL ARCHITECTURE

There are two kinds of models in ADCOL: the local models trained in the parties and the discriminator trained in the server. As ADCOL works from the perspective of representation, the architecture of the local model is similar as existing studies (Chen et al., 2020; Chen & He, 2021) on self-supervised representation learning. The local model has three components: a base encoder, a projection head, and a predictor. The base encoder (e.g., ResNet-50) extracts representation vectors from inputs. Like SimCLR (Chen et al., 2020) and SimSam (Chen & He, 2021), an additional projection head is introduced to map the representation to a space with a fixed dimension. The final predictor is used to output probabilities for each class. For ease of presentation, we use $F(\cdot)$ to denote the whole model and $G(\cdot)$ to denote the model before the predictor (i.e., $G(\mathbf{x})$ is the mapped representation vector of input $\mathbf{x}$). For the discriminator, we simply use a MLP.

### 3.4 THE OVERALL FRAMEWORK

The overall framework is shown in Figure 1 and Algorithm 1. There are four steps in each round: (1) The server sends the discriminator to the parties. (2) The parties update their local models. (3) The parties send representations to the server. (4) The server updates the discriminator.

**Step 1**   In the first step, the server sends the discriminator to parties (line 4 of Algorithm 1).

**Step 2**   In the second step, the parties update their models using their local datasets (lines 10-17 of Algorithm 1). In addition to the objective which aims to minimize the cross-entropy loss (i.e., $\ell_{CE}$) on the local dataset, ADCOL introduces an additional regularization term which aims to maximize the probability that the discriminator cannot distinguish the local representations. For each input $\mathbf{x}$, ADCOL feeds the representation $G(\mathbf{x})$ to the discriminator. ADCOL expects the discriminator to output probability vector $[\frac{1}{N}]^N$ (i.e., the probability of each class is $\frac{1}{N}$) such that it cannot distinguish which party that the representation comes from. Thus, ADCOL uses Kullback–Leibler (KL) divergence loss to measure the difference between the output of the discriminator $D(G(\mathbf{x}))$ and the target $[\frac{1}{N}]^N$. The final loss of an input $(\mathbf{x}, y)$ is computed as

$$\ell = \ell_{CE}(F(\mathbf{x}), y) + \mu \ell_{KL}([\frac{1}{N}]^N \parallel D(G(\mathbf{x}))) \tag{2}$$

where $\mu$ is a hyper-parameter to control the weight of KL divergence loss, $\ell_{CE}$ is the cross-entropy loss, and $\ell_{KL}$ is the KL divergence loss. Each party minimizes its local empirical risk $\mathbb{E}_{(\mathbf{x},y)\sim\mathcal{D}_i}\ell(\mathbf{x}, y; D)$ to update its local model, where $\ell(\cdot)$ is presented in Equation 2.

**Step 3**   After local training, the parties feed their data into the local models and transfer the representations to the server (line 5 of Algorithm 1).

**Step 4**   The server updates the discriminator using the received representations (lines 6-9 of Algorithm 1). Specifically, the server builds a training set $\mathcal{D}_R = \{\mathbf{R}, I\}$, where the feature values are the representations and the labels are the party IDs that the representations come from. The server minimizes the empirical risk $\mathbb{E}_{(\mathbf{R},I)\sim\mathcal{D}_R}\ell_{CE}(\mathbf{R}, I)$ on the training set to update the discriminator.

## 4   THEORETICAL ANALYSIS

### 4.1   CONVERGENCE OF ADCOL

As shown in Equation 2, the local loss has two parts: the cross-entropy loss part to update the whole network $F$ and the KL divergence loss part to update the representation generator $G$. Ideally, to achieve minimum of $\ell$, each part should achieve minimum. Since the cross-entropy loss part is same as FedAvg, we focus on the effect of the KL divergence loss. For simplicity, we ignore the cross-entropy loss and study the KL divergence loss in our theoretical analysis[1]. The local objective

---

[1]Note that $G$ is a part of $F$ and the two losses are not independent of each other. For simplicity, we only analyze the KL divergence loss to study its effect.

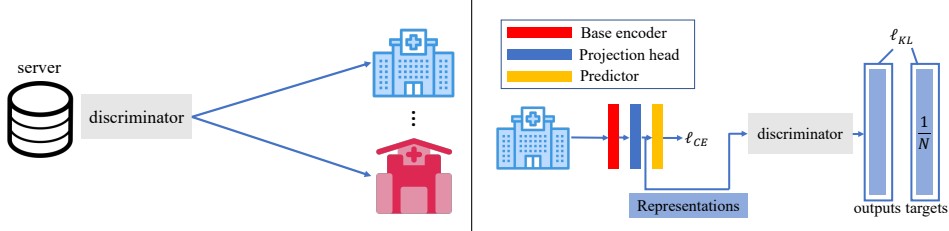

(a) Step 1: The server sends the discriminator to the parties.

(b) Step 2: The parties update their local models

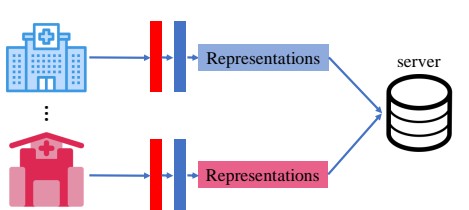

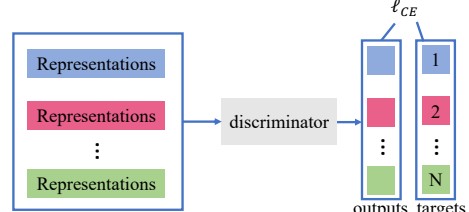

(c) Step 3: The parties send representations to the server.

(d) Step 4: The server updates the discriminator.

Figure 1: The ADCOL framework

---

**Algorithm 1:** The ADCOL algorithm

**Input:** number of communication rounds $T$, number of parties $N$, number of local epochs $E$, learning rate $\eta$, hyper-parameter $\mu$
**Output:** The local models $F_i$ $(i \in [N])$

1 **Server executes:**
2 **for** $t = 1, 2, ..., T$ **do**
3     **for** $i = 1, 2, ..., N$ **in parallel do**
4        send the discriminator $D$ to party $i$
5        $\mathbf{R}_i \leftarrow$ **PartyLocalTraining**$(i, D)$
6     $\mathbf{R} \leftarrow \{(\mathbf{R}_i, i)\}_{i=1}^N$
7     **for** each batch $\mathbf{b} = \{\mathbf{R}_i, i\}$ of $\mathbf{R}$ **do**
8        $\ell \leftarrow CrossEntropyLoss(D(\mathbf{R}_i), i)$
9        $D \leftarrow D - \eta \nabla \ell$

10 **PartyLocalTraining**$(i, D)$:
11 **for** epoch $e = 1, 2, ..., E$ **do**
12     **for** each batch $\mathbf{b} = \{\mathbf{x}, y\}$ of $\mathcal{D}^i$ **do**
13        $\ell_{CE} \leftarrow CrossEntropyLoss(F_i(\mathbf{x}), y)$
14        $\mathbf{R} \leftarrow G_i(\mathbf{x})$
15        $\ell_{KL} \leftarrow KLDiv([\frac{1}{N}]^N || D(\mathbf{R}))$
16        $\ell \leftarrow \ell_{CE} + \mu \ell_{KL}$
17        $F_i \leftarrow F_i - \eta \nabla \ell$
18 **return** $G_i(\mathbf{x}^i)$ to server

---

of party $i$ is:

$$\min_{G_i} \mathbb{E}_{\mathbf{x} \sim \mathcal{D}_i} \ell_{KL}([\frac{1}{N}]^N \, || \, D(G_i(\mathbf{x}))). \tag{3}$$

The objective of the discriminator is

$$\max_D \sum_{i=1}^N \mathbb{E}_{\mathbf{x} \sim \mathcal{D}_i} \log(D_i(G_i(\mathbf{x}))), \tag{4}$$

where $D_i(\cdot)$ is the $i$-th output of the prediction vector $D(\cdot)$ (i.e., the probability of class $i$). Here we analyze the convergence property of the training process like existing studies on GANs (Goodfellow et al., 2014; Tran et al., 2019). In Theorem 4.1, we derive the optimal discriminator given the objective Equation Equation 4. Then, in Theorem 4.2, we derive the optimal solution for the distributions of local representations to minimize the local objective Equation Equation 3 given the optimal discriminator from Theorem 4.1. Last, in Theorem 4.3, we show that the distribution of local representations can converge to optimal solution given in Theorem 4.2. All the proofs are available in Appendix A.

**Theorem 4.1.** *We use $P_{G_i}$ to denote the distribution of the representations generated in party $i$ and $P_{G_i}(\mathbf{z})$ is the probability of representation $\mathbf{z}$ in distribution $P_{G_i}$. Then, the optimal discriminator $D^*$*

*of Equation 4 is*

$$D_k^*(\mathbf{z}) = \frac{P_{G_k}(\mathbf{z})}{\sum_{i=1}^{N} P_{G_i}(\mathbf{z})}. \tag{5}$$

**Theorem 4.2.** *Given the optimal discriminator $D^*$ from Equation 5, the global minimum of Equation 3 is achieved if and only if*

$$P_{G_1} = P_{G_2} = \cdots = P_{G_N} \tag{6}$$

Theorem 4.1 and Theorem 4.2 show that to achieve the minimum of the objectives of the local parties and the discriminator, the parties will generate the same representation distribution, which matches the goal of ADCOL. In Theorem 4.2, we assume that $D$ can reach $D^*$ like existing GAN studies (Goodfellow et al., 2014; Tran et al., 2019). For detailed analysis on it, please refer to Appendix A.

**Theorem 4.3.** *Suppose $P_G^*$ is the optimal solution shown in Theorem 4.2. If $G_i$ ($\forall i \in [1, N]$) and $D$ have enough capacity, and $P_{G_i}$ is updated to minimize the local objective (i.e., Equation 3), given the optimal discriminator $D^*$ from Equation 5, then $P_{G_i}$ converges to $P_G^*$.*

The above theorem provides insights on the convergence of the training. In practice, we optimize the parameter $\theta$ of the local networks rather than $P_{G_i}$ itself, which is reasonable due to the excellent performance as claimed in Goodfellow et al. (2014). Note that there are collapsing solutions for Equation 3 and Equation 4. The representations of each party can simply be constant vectors, which can achieve global minimum of Equation 3. Thus, the cross-entropy loss is necessary in Equation 2, which ensures that the generated representations are meaningful.

## 4.2 COMMUNICATION SIZE

For simplicity, our analysis assumes all parties participate in learning in each round, and it is straightforward to extend this assumption by considering party sampling techniques. We use $S_L$ to denote the size of the local model. Then, the communication size per round of FedAvg is $2NS_L$, including the server sends the model to all parties and the parties send their local models to the server. We use $n$ to denote the total number of examples (i.e., $n = \sum_{i=1}^{N} |\mathcal{D}^i|$), $d$ to denote the dimension of the representations, and $S_D$ to denote the size of the discriminator. Suppose each float value costs four bytes to store. In each round, the communication size of ADCOL is $(4nd + NS_D)$, including the parties send the representations to the server and the server sends the discriminator to the parties.

Although the communication costs of ADCOL and FedAvg depend on the specific settings, we find that ADCOL is usually more communication-efficient than FedAvg in the experimental setting of existing studies. For example, in the experimental setting of FedAvg [5], a simple MLP (199,210 parameters) is used for classification on MNIST with 100 parties. For FedAvg, the communication size per round is $159.4MB$. For ADCOL, considering the dimension of representation $d = 128$, and the discriminator as a 2-layer MLP with 128 hidden units ($S_d = 68,628B$). Then, the communication size per round of ADCOL is $37.6MB$. There may exist extreme cases that ADCOL has more communication size per round than FedAvg when the number of parties is small, the number of samples is large, and the size of model is small. However, in such cases, it is usually impractical to conduct FL as local training may already achieve satisfactory performance.

Like party sampling in FedAvg, we propose **representation sampling** to further reduce the communication size. In each round, each party randomly samples a subset of the representations and sends them to the server. As we will show in Appendix B.11, representation sampling can effectively reduce the communication size with tolerable accuracy loss.

## 4.3 PRIVACY

While sharing representation is used in ADCOL and other collaborative learning studies (He et al., 2020; Peng et al., 2019b; Vepakomma et al., 2018), one possible concern is that representations may leak more information than models. There are many existing studies (Shokri et al., 2017; Nasr et al., 2019) that infer sensitive information from exchanged gradients/models. Also, there are studies (Salem et al., 2020) on reconstruction attacks on the output of a model. While existing studies have shown that the mutual information between the input data and the final representation is small

(Shwartz-Ziv & Tishby, 2017), it is still not clear that whether sharing models is more private than sharing representations to the best of our knowledge, which can be an interesting future direction.

To enhance the privacy guarantee, Differential Privacy (DP) (Dwork et al., 2014) can be applied to protect the transferred messages including the representation. For more details, please refer to Appendix B.12.

## 5 EXPERIMENTS

### 5.1 EXPERIMENTAL SETUP

**Baselines** We compare ADCOL with seven baselines including SOLO (i.e., each party trains the model individually without collaborative learning), FedAvg (McMahan et al., 2016), FedBN (Li et al., 2021c), PartialFed (Sun et al., 2021), FedProx (Li et al., 2020), Per-FedAvg (Fallah et al., 2020), and FedRep (Collins et al., 2021). Here FedBN is the state-of-the-art FL approach on non-IID features. PartialFed is a personalized FL approach on the cross-domain setting which is also applicable to the non-IID feature setting. FedProx is a popular FL approach for non-IID data. Per-FedAvg and FedRep are two state-of-the-art personalized FL approaches. FedUFO is not open-sourced and requires all-to-all communication of local models among any two parties during local training, which leads to prohibitively high communication cost. For example, in our experimental setting with *Digits* task, FedUFO has 77 times higher communication cost than ours. Thus, we omit the experiments with FedUFO here. Like FedAvg (McMahan et al., 2016), we use weighted average according to the data volume of each party for all baselines. By default, we do not apply representation sampling and differential privacy in ADCOL.

**Models** All approaches use the same local model architecture for a fair comparison. The architecture of the local model is similar as SimSam (Chen & He, 2021), which has the following three components: (1) Base encoder: ResNet-50 (He et al., 2016). (2) Projection head: a 3-layer MLP with BN applied to each fully-connected layer. The input dimension is 4096. The dimension of the hidden layer and the output layer is 2048. (3) Predictor: a 2-layer MLP with BN applied to its hidden layer. The input dimension is 2048. The dimension of its hidden layer is 512. The discriminator is a 3-layer MLP. The input dimension is 2048. The dimension of the hidden layers is 512. The output dimension is equal to the number of parties.

**Datasets** We use the same tasks as in the study of FedBN. There are three tasks in our experiments: (1) *Digits*: The Digits task has the following five digit data sources from different domains: MNIST (LeCun et al., 1998), SVHN (Netzer et al., 2011), USPS (Hull, 1994), SynthDigits (Ganin & Lempitsky, 2015), and MNIST-M (Ganin & Lempitsky, 2015). (2) *Office-Caltech-10* (Gong et al., 2012): The dataset has four data sources acquired using different camera devices or in different real environments with various backgrounds: Amazon, Caltech, DSLR, and WebCam. (3) *DomainNet* (Peng et al., 2019a): The dataset contains natural images coming from six different data sources with different image styles: Clipart, Infograph, Painting, Quickdraw, Real, and Sketch. Here the first task is a synthetic task by combining different digit datasets. The second and third tasks are real-world datasets that naturally generated in a federated setting. For each task, different datasets have heterogeneous features but share the same label distribution, which naturally forms the non-IID feature setting (Li et al., 2021a). Due to the page limit, we only present some experimental results on Digits in the main paper. For more experimental results and details, please refer to Appendix B.

**Setup** By default, the number of parties is equal to the number of data sources, where each party has data from one of the data sources. For each dataset, we randomly split 1/5 of the original dataset as the test dataset, while the remained dataset is used as the training dataset. The number of local epochs is set to 10 by default for all FL approaches. The number of epochs is set to 300 for SOLO. For ADCOL and FedProx, we tune $\mu \in \{10, 1, 0.1, 0.01, 0.001\}$ and report the best results. For FedRep, we tune $\beta$ (i.e., step size for the second batch training) from $\{0.001, 0.01\}$ and report the best results. We use the prediction layers as the shared representation in FedRep. We use PyTorch (Paszke et al., 2019) to implement all approaches. We use the SGD optimizer for training with a learning rate of 0.01. The SGD weight decay is set to $10^{-5}$ and the SGD momentum is set to 0.9. The batch size is set to 64, 32, and 32 for Digits, Office-Caltech-10, and DomainNet, respectively. We run the

Table 1: The comparison of top-1 test accuracy among different approaches on Digits. We run FL approaches for 100 rounds (all approaches have converged). We run three trials and report the mean and standard derivation. Besides the test accuracy on each party, we also report the mean accuracy of all parties denoted as "AVG".

| Digits | MNIST | SVHN | USPS | SynthDigit | MNIST_M | AVG |
|---|---|---|---|---|---|---|
| SOLO | 87.9%±0.4% | 34.8%±0.8% | 94.8%±0.1% | 63.0%±0.4% | 67.2%±0.4% | 69.5%±0.3% |
| FedAvg | 94.4%±0.5% | 59.4%±0.9% | 94.3%±0.2% | 74.4%±0.5% | 70.3%±1.2% | 78.6%±0.6% |
| FedBN | 94.1%±0.8% | **59.9**%±0.7% | 94.1%±0.1% | 73.9%±0.6% | 71.3%±1.1% | 78.7%±0.6% |
| PartialFed | **94.7**%±0.4% | 59.4%±0.6% | 94.2%±0.1% | 75.2%±0.4% | 69.7%±0.6% | 78.6%±0.4% |
| FedProx | 94.1%±0.4% | 59.8%±0.6% | 94.3%±0.1% | 73.4%±0.3% | 71.6%±0.9% | 78.6%±0.4% |
| Per-FedAvg | 88.9%±0.7% | 36.6%±1.3% | 89.5%±0.2% | 58.3%±0.7% | 54.5%±1.3% | 65.6%±0.8% |
| FedRep | 92.6%±0.2% | 42.0%±1.0% | 93.1%±0.1% | 61.1%±0.5% | 50.8%±1.4% | 67.9%±0.8% |
| ADCOL | **94.7**%±0.6% | 58.2%±1.0% | **95.4**%±0.2% | **76.0**%±0.3% | **76.7**%±0.8% | **80.2**%±0.5% |

experiments on a server with 8 * NVIDIA GeForce RTX 3090, a server with 4 * NVIDIA A100, and a cluster with 45 * NVIDIA GeForce RTX 2080 Ti.

## 5.2 OVERALL COMPARISON

Table 1 reports the test accuracy of different approaches on three tasks. We have the following observations. First, ADCOL is more effective than the other approaches. It can achieve the best test accuracy on most datasets. Moreover, ADCOL can outperform the other approaches by more than 2% accuracy on average. Second, while the parties may not benefit from FL approaches in some cases (e.g., Caltech-10), ADCOL always achieves better accuracy than SOLO, which demonstrates the robustness of ADCOL. Last, the personalized FL approaches (i.e., Per-FedAvg and FedRep) have a poor performance on the non-IID feature setting, which are even worse than SOLO. For the results of other tasks, please refer to Appendix B.2.

## 5.3 COMMUNICATION EFFICIENCY

To show the communication efficiency of ADCOL, like existing studies (Karimireddy et al., 2020; Lin et al., 2020), we compare the number of communication rounds and communication size of each approach to achieve the same target performance. The results on Digits are shown in Table 2. We can observe that no approach consistently outperforms the other approaches in terms of the number of communication rounds. However, the communication size of ADCOL is always much smaller than the other approaches. ADCOL can save at least 10 times the communication costs to achieve the same accuracy as FedAvg. The speedup can even be up to 34× on Digits. The results demonstrate that ADCOL is much more communication-efficient than the other FL approaches. For the results of other tasks, please refer to Appendix B.2.

## 5.4 SENSITIVITY STUDIES

**Scalability and Heterogeneity**   We adopt the same approach as Li et al. (2021c) to study the effect of number of parties and heterogeneity. We divide each dataset into ten parts randomly and equally and allocate each part into one party. The parties from the same dataset are treated as IID and the parties from different datasets are treated as non-IID. We add two parties from each dataset each time, which results in the number of parties $N \in \{10, 20, 30, 40, 50\}$. Moreover, the degree of heterogeneity decreases as the number of parties increases since the number of IID parties increases. The test accuracies are reported in Figure 2a. We can observe that the accuracy of all approaches can be slightly improved when increasing the number of parties due to the reduced heterogeneity and increased total amount of data. Given a different number of parties, ADCOL consistently outperforms the other baselines. Although the number of classes to distinguish increases for the discriminator when increasing $N$, ADCOL still shows a good and stable performance.

**Effect of Local Dataset Size**   We vary the percentage of the original local dataset used in each party from 20% to 100%. The results are shown in Figure 2b. The improvement of ADCOL is more significant when the size of the local dataset is small. If the size of the local dataset is large, each party can already achieve satisfactory accuracy by SOLO. The accuracy of all approaches is close when the percentage is 100%. It is not necessary to conduct collaborative learning in such a case.

Table 2: The communication round and size of each approach to achieve the same target performance as the minimum converged accuracy among FedAvg, FedBN, PartialFed, FedProx, and ADCOL as shown in Table 1 (i.e., 94.1% in MNIST). We use the slash cell to indicate that the approach (i.e., Per-FedAvg and FedRep) cannot reach the target performance in 100 rounds/30 GB. The speedup is computed by dividing the communication size of FedAvg by the communication size of ADCOL.

| Digits | | MNIST | SVHN | USPS | SynthDigit | MNIST_M | AVG |
|---|---|---|---|---|---|---|---|
| #communication round | FedAvg | 11 | 54 | 5 | 11 | 7 | 28 |
| | FedBN | 11 | 73 | 5 | 68 | 7 | 22 |
| | PartialFed | 9 | 23 | 6 | 14 | 8 | 14 |
| | FedProx | 64 | 42 | 8 | 12 | 10 | 31 |
| | Per-FedAvg | ╲ | ╲ | ╲ | ╲ | ╲ | ╲ |
| | FedRep | ╲ | ╲ | ╲ | ╲ | ╲ | ╲ |
| | ADCOL | 19 | 86 | 6 | 19 | 9 | 21 |
| communication size (GB) | FedAvg | 3.12 | 15.34 | 1.42 | 3.12 | 1.99 | 7.95 |
| | FedBN | 3.12 | 20.73 | 1.42 | 19.31 | 1.99 | 6.25 |
| | PartialFed | 2.56 | 6.53 | 1.70 | 3.98 | 2.27 | 3.98 |
| | FedProx | 18.18 | 11.93 | 2.27 | 3.41 | 2.84 | 8.80 |
| | Per-FedAvg | ╲ | ╲ | ╲ | ╲ | ╲ | ╲ |
| | FedRep | ╲ | ╲ | ╲ | ╲ | ╲ | ╲ |
| | ADCOL | 0.21 | 0.95 | 0.07 | 0.21 | 0.10 | 0.23 |
| Speedup | | **14.95x** | **16.21x** | **21.52x** | **14.95x** | **20.08x** | **34.42x** |

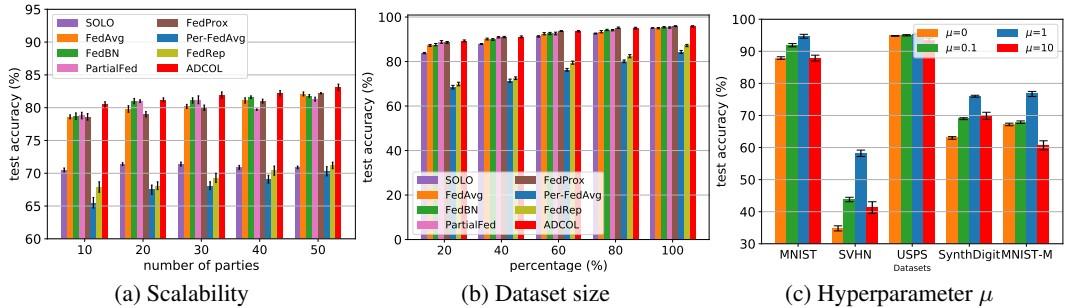

(a) Scalability      (b) Dataset size      (c) Hyperparameter $\mu$

Figure 2: Effect of different factors. We run three trials and report the mean accuracy across parties and its standard derivation.

**Effect of $\mu$** We vary $\mu \in \{0, 0.1, 1, 10\}$ and report the accuracy of ADCOL as shown in Figure 2c. We can observe that ADCOL can achieve the best accuracy when $\mu = 1$. If $\mu$ is too small, the KL divergence loss of Equation 2 has little effect on the local training. Then, the goal of learning a common representation distribution may not achieve. If $\mu$ is too large, the cross-entropy loss of Equation 2 has little effect on the local training, and the representations may not be useful for classification at all (e.g., all representations are a constant vector). ADCOL with $\mu = 10$ may even be worse than SOLO (i.e., $\mu = 0$). Thus, an appropriate $\mu$ is important in ADCOL. Through our experimental studies, we find that setting $\mu = 1$ is a good default choice.

## 6 CONCLUSION

In this paper, we propose ADCOL, a novel collaborative learning approach for non-IID features. Unlike most previous studies performing model averaging, ADCOL trains the models in an adversarial way between the parties and the server from the perspective of representation distributions. The parties aim to learn a common representation distribution, while the server aims to distinguish the representations by party IDs. Our experiments on three real-world tasks show that ADCOL achieves higher accuracy than the other state-of-the-art federated learning approaches on non-IID features.

ADCOL shows that it is possible to incorporate global knowledge into parties in an adversarial way instead of model averaging. This is a fundamentally new and potentially powerful way for federated learning. We are interested in future studies on extending ADCOL to more federated settings and advanced techniques for efficient and privacy-preserving representation sharing.

**Reproducibility Statement**   We have provided the experimental details in Section 5.1 and Appendix B.1 for reproducibility. Moreover, we will make the code publicly available.

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

# A    THEORETICAL ANALYSIS

**Theorem 4.1.** *We use $P_{G_i}$ to denote the distribution of the representations generated in party $i$ and $P_{G_i}(\mathbf{z})$ is the probability of representation $\mathbf{z}$ in distribution $P_{G_i}$. Then, the optimal discriminator $D^*$ of Equation 4 of the main paper is*

$$D_k^*(\mathbf{z}) = \frac{P_{G_k}(\mathbf{z})}{\sum_{i=1}^N P_{G_i}(\mathbf{z})}. \tag{7}$$

*Proof.* From the view of the distribution of representations $\mathbf{z}$, we can reformulate Equation 4 and the objective is to maximize:

$$\sum_{i=1}^N \int_{\mathbf{z}} P_{G_i}(\mathbf{z}) \log(D_i(\mathbf{z})) d\mathbf{z} \tag{8}$$

Let $V(D) = \sum_{i=1}^N P_{G_i}(\mathbf{z}) \log(D_i(\mathbf{z}))$. To maximize Equation 8 with respect to $D$, it is equivalent to maximize $V(D)$ with respect to $D$ given any $z$. Note that $\sum_{i=1}^N D_i(\mathbf{z}) = 1$. Let $F(D) = V(D) + \lambda(1 - \sum_{i=1}^N D_i(\mathbf{z}))$. We have

$$\frac{\partial F(D)}{\partial D_i(\mathbf{z})} = \frac{P_{G_i}(\mathbf{z})}{D_i(\mathbf{z})} - \lambda \tag{9}$$

Let $\frac{\partial F(D)}{\partial D_i(\mathbf{z})} = 0$ for $i \in [1, N]$, we have

$$\frac{P_{G_1}(\mathbf{z})}{D_1(\mathbf{z})} = \frac{P_{G_2}(\mathbf{z})}{D_2(\mathbf{z})} = \cdots = \frac{P_{G_N}(\mathbf{z})}{D_N(\mathbf{z})} = \lambda \tag{10}$$

Thus, $V(D)$ can achieve maximum when

$$D_k^*(\mathbf{z}) = \frac{P_{G_k}(\mathbf{z})}{\sum_{i=1}^N P_{G_i}(\mathbf{z})} \tag{11}$$

$\square$

Note that the discriminator uses SGD to update its model with cross-entropy loss as shown in Lines 6-9 of Algorithm 1. If the discriminator is a linear function, then it can converge to the global optima since the loss function is convex. If the discriminator is a neural network with non-linear activations, whether SGD finds a global minimum or not is a traditional optimization problem, which is orthogonal to our study. Given the evidence of the power of deep learning with SGD from existing studies (Du et al., 2019; Zhou et al., 2019; Zou et al., 2018; Choromanska et al., 2015; Dauphin et al., 2014), we assume that $D$ can reach $D^*$ like existing GAN studies (Goodfellow et al., 2014; Tran et al., 2019). We also empirically show that the discriminator can converge to optima in Appendix B.15.

**Theorem 4.2.** *Given the optimal discriminator $D^*$ from Equation 7, the global minimum of Equation 3 of the main paper is achieved if and only if*

$$P_{G_1} = P_{G_2} = \cdots = P_{G_N} \tag{12}$$

*Proof.* From Equation 3 of the main paper, the local objective of party $k$ is to minimize

$$
\begin{aligned}
W(G_k) &= -\mathbb{E}_{\mathbf{x} \sim \mathcal{D}_k} \frac{1}{N} \sum_{i=1}^N \log(N \cdot D_i(G_k(\mathbf{x}))) \\
&= -\mathbb{E}_{\mathbf{x} \sim D_k} \frac{1}{N} \sum_{i=1}^N \log\left(\frac{N \cdot P_{G_i}(G_k(\mathbf{x}))}{\sum_{j=1}^N P_{G_j}(G_k(\mathbf{x}))}\right) \\
&= -\mathbb{E}_{\mathbf{x} \sim D_k} \frac{1}{N} \sum_{i=1}^N \left(\log N + \log\left(\frac{P_{G_i}(G_k(\mathbf{x}))}{\sum_{j=1}^N P_{G_j}(G_k(\mathbf{x}))}\right)\right)
\end{aligned} \tag{13}
$$

To minimize Equation 13, we need to maximize $\log(\frac{P_{G_i}(G_k(\mathbf{x}))}{\sum_{j=1}^N P_{G_j}(G_k(\mathbf{x}))})$. Note that $\sum_{i=1}^N \frac{P_{G_i}(G_k(\mathbf{x}))}{\sum_{j=1}^N P_{G_j}(G_k(\mathbf{x}))} = 1$. Similar to the proof in Theorem 4.1, Equation 13 can achieve minimum when

$$\frac{P_{G_1}(G_k(\mathbf{x}))}{\sum_{j=1}^N P_{G_j}(G_k(\mathbf{x}))} = \frac{P_{G_2}(G_k(\mathbf{x}))}{\sum_{j=1}^N P_{G_j}(G_k(\mathbf{x}))} = \cdots = \frac{P_{G_N}(G_k(\mathbf{x}))}{\sum_{j=1}^N P_{G_j}(G_k(\mathbf{x}))} = \frac{1}{N}. \tag{14}$$

For $\forall k \in [1, N]$ and $\forall i \in [1, N]$, we have $P_{G_i}(G_k(\mathbf{x})) = \frac{\sum_{j=1}^N P_{G_j}(G_k(\mathbf{x}))}{N}$. Given a representation $\mathbf{z}$, we have

$$P_{G_1}(\mathbf{z}) = P_{G_2}(\mathbf{z}) = \cdots = P_{G_N}(\mathbf{z}) = \frac{\sum_{j=1}^N P_{G_j}(\mathbf{z})}{N} \tag{15}$$

Thus, $P_{G_1} = P_{G_2} = \cdots = P_{G_N}$. $\qquad\square$

**Theorem 4.3.** *Suppose $P_G^*$ is the optimal solution shown in Theorem 4.2. If $G_i$ ($\forall i \in [1, N]$) and $D$ have enough capacity, and $P_{G_i}$ is updated to minimize the local objective (i.e., Equation 3 of the main paper), given the optimal discriminator $D^*$ from Equation 7, then $P_{G_i}$ converges to $P_G^*$.*

*Proof.* In Equation 13, consider $W(G_k) = U(P_{G_i})$ as a function of $P_{G_i}$. Then

$$\frac{\partial U(P_{G_i})}{\partial P_{G_i}} = -\frac{\sum_{k \neq i} P_{G_k}}{N P_{G_i}(P_{G_i} + \sum_{k \neq i} P_{G_k})}. \tag{16}$$

We have

$$\frac{\partial^2 U(P_{G_i})}{\partial P_{G_i}^2} = \frac{1}{P_{G_i}^2} - \frac{1}{(P_{G_i} + \sum_{k \neq i} P_{G_k})^2} \geq 0 \tag{17}$$

Thus, $U(P_{G_i})$ is convex in $P_{G_i}$. Therefore, with sufficiently small updates of $P_{G_i}$, $P_{G_i}$ converges to $P_G^*$, concluding the proof. $\qquad\square$

## B   ADDITIONAL EXPERIMENTAL RESULTS

### B.1   ADDITIONAL EXPERIMENTAL DETAILS

In each experiment, like FedBN (Li et al., 2021c), to remove the effect of quantity skew, we truncate the size of all datasets to their smallest number with random sampling. For Digits, we resize all images to $28 \times 28 \times 3$ and normalize them with mean 0.5 and standard derivation 0.5 for each channel. For Office-Caltech-10, we resize all images to $64 \times 64 \times 3$ with random horizontal flip and random rotation. For DomainNet, we resize all images to $64 \times 64 \times 3$ with random horizontal flip and random rotation. Like FedBN, we take Digits as the benchmark task for most studies.

The statistics of all the datasets are shown in Table 3. To quantitatively demonstrate the feature imbalance, we use FID (Heusel et al., 2017) to measure the difference between feature distributions of different parties. Specifically, it measures the Frechet distance between the representation distributions of different datasets, where the representation is generated by a Inception v3 model pretrained on ImageNet dataset. FID is 0 when two datasets are the same. For each task, we compute the FID between each subset and the whole dataset by merging all subsets. With FID values for each subset, we report the mean value and the standard deviation. From Table 3, we can observe that there indeed exists feature imbalance for each task. We also show the label distributions in Figure 3. The portion of samples with each class is close to 0.1. The label distribution is balanced among the parties. We train a ResNet-50 on all datasets (i.e., parties) from a task and extract the feature distributions of each dataset. Then, we use t-SNE to visualize the representation as shown in Figure 4. We can observe that the feature distribution of each party is different.

There are two major differences between our experimental setup and the setup in FedBN. (1) The model architecture is different. Our paper adopts ResNet-50 for all datasets, while FedBN uses a simple CNN for Digit and AlexNet for Office and DomainNet. We adopt ResNet since we need to

Table 3: The statistics of all studied datasets.

|  |  | #training samples | #testing samples | FID |
|---|---|---|---|---|
| Digits | MNIST | 56,000 | 14,000 | 140.97±30.61 |
|  | SVHN | 79,431 | 19,858 |  |
|  | USPS | 7,438 | 1,860 |  |
|  | SynthDigit | 402,209 | 97,791 |  |
|  | MNIST_M | 56,000 | 14,000 |  |
| Caltech-10 | Amazon | 766 | 192 | 78.12±43.11 |
|  | Caltech | 898 | 225 |  |
|  | DSLR | 125 | 32 |  |
|  | WebCam | 236 | 59 |  |
| DomainNet | Clipart | 2,103 | 526 | 144.81±35.40 |
|  | Infograph | 2,626 | 657 |  |
|  | Painting | 2,472 | 619 |  |
|  | Quickdraw | 4,000 | 1,000 |  |
|  | Real | 4,864 | 1,217 |  |
|  | Sketch | 2,213 | 554 |  |

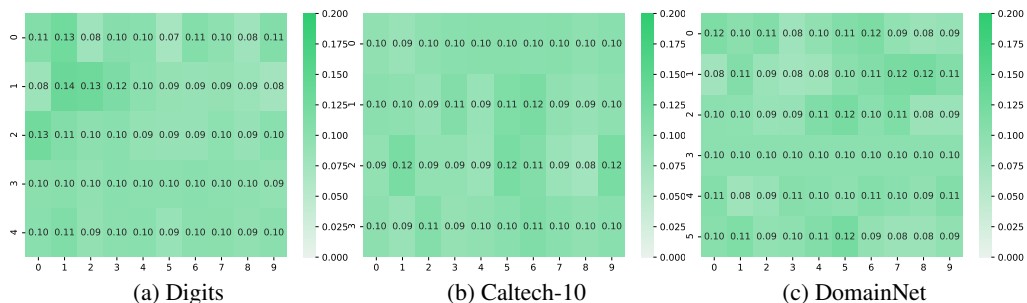

(a) Digits  (b) Caltech-10  (c) DomainNet

Figure 3: The label distributions of each task. The value in each cell of row $i$ and column $j$ represents the percentage of samples with class $j$ in Party $i$.

extract the representations for the input data, and ResNet-50 is commonly used as a base encoder to extract the representations in representation learning studies (Chen et al., 2020; Grill et al., 2020; Chen & He, 2021). (2) The image size is different. FedBN resizes all images of Office and Domain to 256x256x3, while our paper resizes them to 64x64x3 for computation efficiency.

## B.2 CALTECH-10 AND DOMAINNET

Table 4 and 5 show the test accuracy of different approaches on Caltech-10 and DomainNet, respectively. We can observe that ADCOL still outperforms the other approaches in most cases.

We show the communication efficiency of ADCOL on Caltech-10 and DomainNet in Table 6 and Table 7. We can observe that ADCOL is much more communication-efficient than the other approaches. The speedup can be even up to 300 times.

## B.3 EXPLANATION OF THE EXPERIMENTAL RESULTS BY FID

We observe that there is a correlation between FID and the performance gain of ADCOL compared with local training. Generally, with a higher FID (i.e., more imbalanced feature distribution), the party can gain more from our approach. The relative improvement on the accuracy of ADCOL against local training is 15.4%, 6.8%, and 17.2% on Digits, Caltech-10, and DomainNet, respectively. The improvement is positively related to the FID of each task. If FID is small, the representation distribution of the local dataset is close to the global dataset, then local training may already learn a good representation and the improvement of ADCOL is limited.

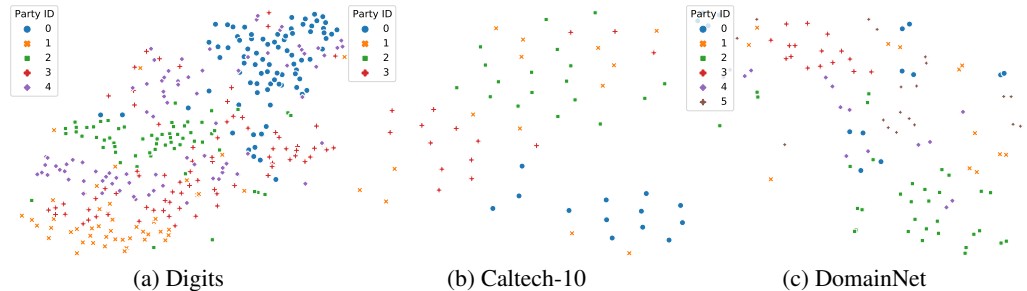

(a) Digits                    (b) Caltech-10                    (c) DomainNet

Figure 4: The feature distributions of each task.

Table 4: The comparison of top-1 test accuracy among different approaches on Caltech-10.

| Caltech-10 | Amazon | Caltech | DSLR | WebCam | AVG |
|---|---|---|---|---|---|
| SOLO | 52.8%±0.9% | 36.0%±0.9% | 71.9%±0.5% | 74.6%±0.5% | 58.8%±0.6% |
| FedAvg | 24.0%±1.7% | 36.9%±1.4% | 81.3%±0.4% | 82.7%±0.4% | 56.2%±1.0% |
| FedBN | 33.3%±1.5% | 33.8%±1.8% | 81.3%±0.5% | 80.0%±0.6% | 57.1%±1.0% |
| PartialFed | 17.4%±2.2% | 24.2%±1.6% | 70.3%±3.4% | 77.1%±3.6% | 47.2%±2.4% |
| FedProx | 43.2%±1.6% | 33.1%±0.8% | **82.6%**±0.4% | **83.1%**±0.7% | 60.5%±0.7% |
| Per-FedAvg | 33.9%±1.6% | 32.4%±1.5% | 62.5%±0.8% | 74.6%±0.8% | 50.8%±1.3% |
| FedRep | 16.1%±1.9% | 22.7%±1.7% | 56.3%±1.0% | 57.6%±1.1% | 38.2%±1.4% |
| ADCOL | **54.2%**±1.1% | **38.2%**±1.3% | 75%±0.6% | **83.1%**±0.5% | **62.6%**±0.9% |

## B.4 TRAINING CURVES

The training curves of different approaches on Digit are shown in Figure 5. We can observe that ADCOL is much more communication-efficient than the other approaches. ADCOL can convergence with a much smaller communication size than the other approaches.

## B.5 PARTY SAMPLING

Party sampling is a technique usually used in the cross-device setting, where a fraction of parties is sampled to participate in federated learning in each round. Here we set the sample fraction to 0.4 in Digit and choose FedAvg and FedBN as the baselines. The training curves are shown in Figure 6. We can observe that all approaches have an unstable accuracy during training due to sampling. Moreover, FedAvg and FedBN have a very poor accuracy, which shows that existing federated learning approaches cannot well support party sampling on non-IID features. ADCOL significantly outperforms the other approaches.

We increase the number of parties to 100 (i.e., divide each dataset to 20 subsets) and vary the sampling rate from $\{0.1, 0.2, 0.5, 1\}$. We run all approaches for 200 rounds. The results are shown in Table 8. We can observe that when the sampling rate decreases, the performance of all approaches decreases. Moreover, the training is more unstable if the sampling rate is smaller. However, ADCOL still significantly outperforms FedAvg and FedBN. It is still a challenging task to develop effective algorithm on the cross-device setting with a low sampling rate.

## B.6 STUDY ON THE DISCRIMINATOR

One natural question is how to increase the information contained in the discriminator to improve the performance of ADCOL. We have tried two approaches.

**Changing Model Architecture**   One approach is to increase the capacity of the discriminator. We change the model architecture to ResNet-50. The results are shown in Table 9. The performance of ADCOL cannot be improved by increasing the capacity of the discriminator.

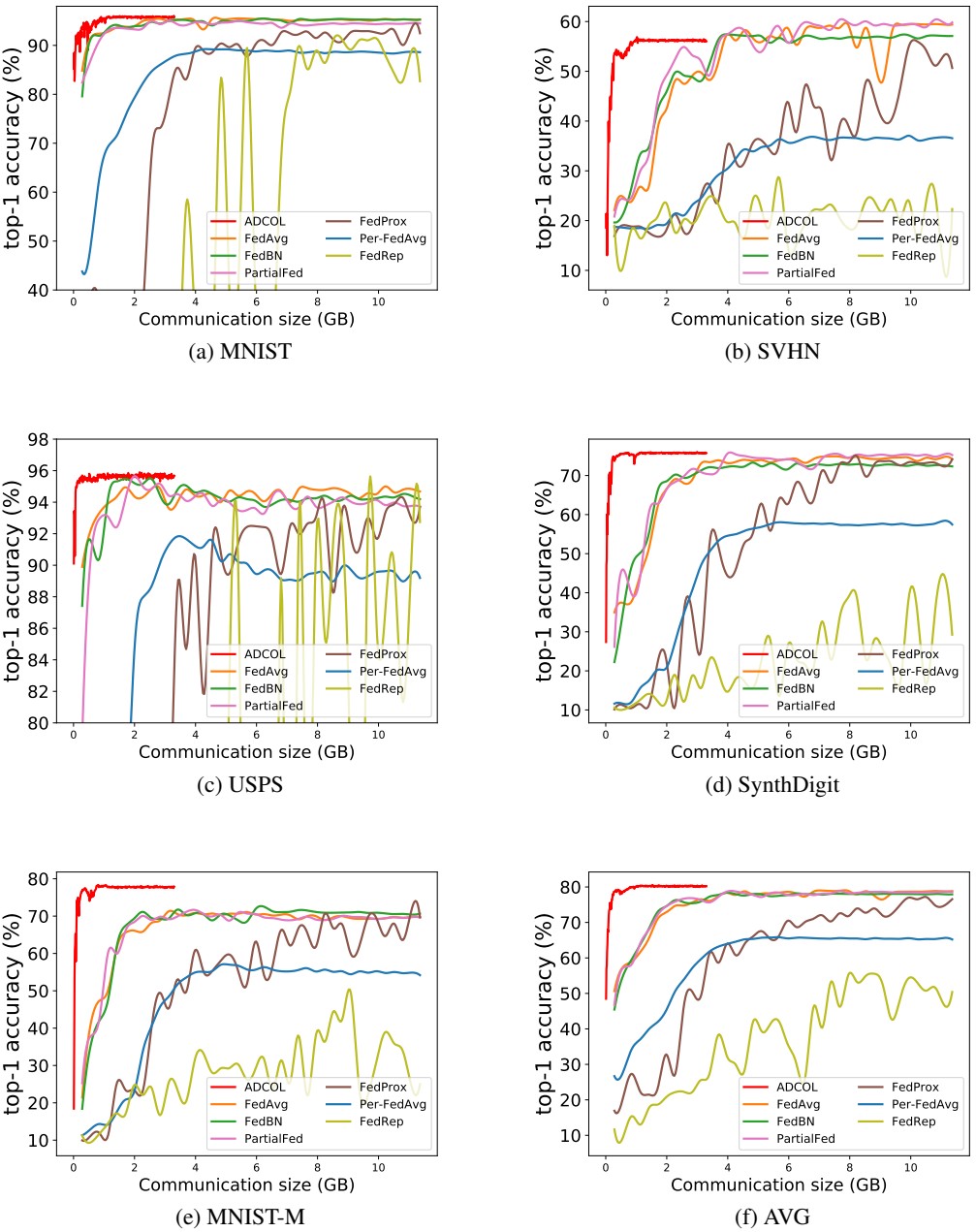

Figure 5: The training curves of different approaches on Digit.

Table 5: The comparison of top-1 test accuracy among different approaches on DomainNet.

| DomainNet | Clipart | Infograph | Painting | Quickdraw | Real | Sketch | AVG |
|---|---|---|---|---|---|---|---|
| SOLO | 31.7%±0.9% | 20.2%±1.2% | 30.9%±0.9% | 48.2%±1.1% | 36.5%±0.8% | 20.8%±1.5% | 31.4%±0.9% |
| FedAvg | 33.5%±1.2% | 20.4%±0.7% | 29.2%±0.8% | 56.2%±1.2% | 40.5%±0.8% | 22.6%±1.1% | 33.7%±0.9% |
| FedBN | 36.3%±1.3% | 20.4%±0.8% | 27.8%±1.4% | 61.3%±0.9% | 41.9%±1.3% | 23.6%±1.2% | 35.2%±1.1% |
| PartialFed | 35.0%±0.2% | 20.5%±0.2% | 30.4%±0.2% | 61.4%±0.7% | 38.3%±2.1% | **27.1%**±0.5% | 35.4%±0.3% |
| FedProx | 37.8%±1.0% | 21.6%±0.8% | 28.1%±1.0% | 23.6%±0.7% | **43.6%**±1.2% | 22.4%±1.0% | 29.5%±1.0% |
| Per-FedAvg | 38.2%±0.7% | 20.2%±0.6% | 27%±1.2% | 42.4%±0.8% | 40.3%±0.7% | 22.6%±1.2% | 31.8%±1.3% |
| FedRep | 27.9%±1.2% | 19%±0.9% | 24.1%±1.2% | 17.9%±1.1% | 31.1%±1.4% | 16.8%±1.1% | 22.8%±1.4% |
| ADCOL | **39.9%**±1.0% | **21.9%**±0.9% | **33.9%**±1.2% | **61.7%**±0.8% | 39.3%±1.5% | 23.9%±1.2% | **36.8%**±1.0% |

Table 6: The communication round and communication cost of each approach to achieve the same target performance on Caltech-10.

| Caltech-10 | | Amazon | Caltech | DSLR | WebCam | AVG |
|---|---|---|---|---|---|---|
| #round | FedAvg | 7 | 22 | 12 | 14 | 14 |
| | FedBN | 3 | 21 | 12 | 19 | 19 |
| | PartialFed | ╲ | ╲ | ╲ | ╲ | ╲ |
| | FedProx | 14 | 29 | 15 | 29 | 22 |
| | Per-FedAvg | 23 | ╲ | ╲ | ╲ | ╲ |
| | FedRep | ╲ | ╲ | ╲ | ╲ | ╲ |
| | ADCOL | 3 | 18 | 31 | 12 | 20 |
| cost (GB) | FedAvg | 1.99 | 6.25 | 3.41 | 3.98 | 3.98 |
| | FedBN | 0.85 | 5.96 | 3.41 | 5.40 | 5.40 |
| | PartialFed | ╲ | ╲ | ╲ | ╲ | ╲ |
| | FedProx | 3.98 | 8.24 | 4.26 | 8.24 | 6.25 |
| | Per-FedAvg | 6.53 | ╲ | ╲ | ╲ | ╲ |
| | FedRep | ╲ | ╲ | ╲ | ╲ | ╲ |
| | ADCOL | 0.02 | 0.10 | 0.17 | 0.07 | 0.11 |
| Speedup | | **120.48** | **63.11** | **19.99** | **60.24** | **36.15** |

**Increasing the Number of Discriminators**    The other one approach is to increase the number of discriminators. Suppose the number of discriminators is $N_d$ and the current round is $t$. Then, we use discriminators from round $\max(1, t - N_d)$ to round $(t - 1)$ in the local training. The KL divergence loss is computed as

$$\ell = \frac{1}{N_d} \sum_{i=1}^{N_d} \ell_{KL}([\frac{1}{N}]^N \parallel D^i(G(\mathbf{x}))), \tag{18}$$

where $D^i$ is the discriminator trained in round $\max(1, t - i)$. The results are shown in Table 10. ADCOL cannot benefit from more discriminators. When the number of discriminators is larger, the accuracy of ADCOL is even worse. It is a future work to investigate how to integrate more useful information into the discriminator.

### B.7    DIMENSION OF REPRESENTATIONS

Same as SimSam (Chen & He, 2021), we set the dimension of representations (i.e., the output dimension of the projection head, the input dimension of the discriminator) to 2048 by default. As shown in Table 11, we report the performance of ADCOL varying the representation dimension. ADCOL can benefit from a larger representation dimension, where the representations are more informative. The mean accuracy can be improved by about 5% by increasing the dimension from 512 to 2048.

### B.8    NON-IID LABELS

We test the performance of ADCOL on non-IID label settings. Specifically, we sample $p_k \sim Dir_N(0.5)$ and allocate a $p_{k,j}$ proportion of the instances of class $k$ to party $j$, where $Dir(0.5)$ is the Dirichlet distribution with a concentration parameter 0.5. The results are shown in Table 12. ADCOL cannot achieve a better performance than FedAvg and FedBN. Intuitively, the task-specific representations of images from different classes should be very different. If the label distribution

Table 7: The communication round and communication cost of each approach to achieve the same target performance on DomainNet.

| DomainNet | | Clipart | Infograph | Painting | Quickdraw | Real | Sketch | AVG |
|---|---|---|---|---|---|---|---|---|
| #round | FedAvg | 22 | 5 | 9 | 38 | 23 | 17 | 47 |
| | FedBN | 14 | 7 | 5 | 54 | 29 | 18 | 41 |
| | PartialFed | 15 | 4 | 28 | 36 | 15 | 13 | 20 |
| | FedProx | 13 | 8 | 16 | ╲ | 32 | 12 | ╲ |
| | Per-FedAvg | 31 | 19 | 34 | ╲ | 64 | 57 | ╲ |
| | FedRep | ╲ | ╲ | ╲ | ╲ | ╲ | ╲ | ╲ |
| | ADCOL | 11 | 4 | 9 | 6 | 44 | 19 | 26 |
| cost (GB) | FedAvg | 6.25 | 1.42 | 2.56 | 10.79 | 6.53 | 4.83 | 13.35 |
| | FedBN | 3.98 | 1.99 | 1.42 | 15.34 | 8.24 | 5.11 | 11.64 |
| | PartialFed | 4.26 | 1.14 | 7.96 | 10.23 | 2.46 | 3.69 | 5.69 |
| | FedProx | 3.69 | 2.27 | 4.54 | ╲ | 9.09 | 3.41 | ╲ |
| | Per-FedAvg | 8.80 | 5.40 | 9.66 | ╲ | 18.18 | 16.19 | ╲ |
| | FedRep | ╲ | ╲ | ╲ | ╲ | ╲ | ╲ | ╲ |
| | ADCOL | 0.06 | 0.02 | 0.05 | 0.04 | 0.26 | 0.11 | 0.15 |
| Speedup | | **96.9** | **60.6** | **48.5** | **306.9** | **25.3** | **43.4** | **87.6** |

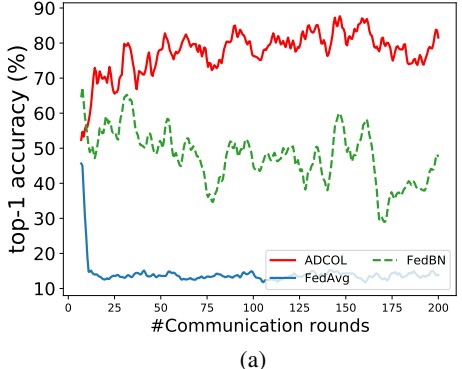
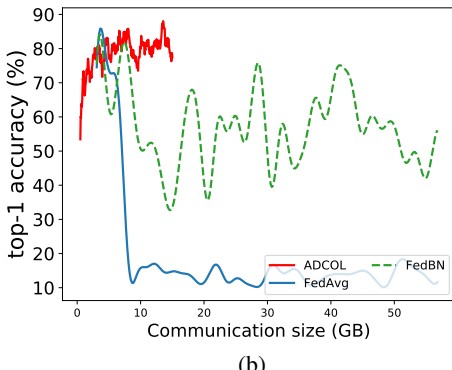

| (a) | (b) |
|---|---|

Figure 6: The training curves with party sampling (sample fraction = 0.4). We report the mean test accuracy across all parties.

varies across parties, the representation distribution naturally also varies a lot. The intuition of ADCOL, which aims to learn a common representation distribution, is not appropriate on non-IID label settings.

## B.9    COMPUTATION OVERHEAD

As shown in Table 13, the training time of ADCOL is larger than the other approaches. ADCOL requires the training of a discriminator in the server side, while the other approaches only need to average the models in the server side. However, in practice, the server usually has much power-

Table 8: The performance of different approaches varying the sampling rate. We run all approaches for 200 rounds and report the final mean accuracy and standard deviation with three runs.

| Sampling Rate | FedAvg | FedBN | ADCOL |
|---|---|---|---|
| 0.1 | 12.1%±9.2% | 25.3%±8.1% | **34.1%±6.5%** |
| 0.2 | 15.3%±7.6% | 31.9%±7.4% | **46.7%±4.2%** |
| 0.5 | 60.2%±5.4% | 62.2%±4.9% | **67.3%±2.4%** |
| 1 | 77.6%±1.4% | 77.7%±1.2% | **79.1%±0.9%** |

Table 9: ADCOL with different discriminator architectures.

| Discriminator | MNIST | SVHN | USPS | SynthDigit | MNIST-M | AVG |
|---|---|---|---|---|---|---|
| ResNet-50 | $95.1\% \pm 0.5\%$ | $55.6\% \pm 0.8\%$ | $96.0\% \pm 0.3\%$ | $73.6\% \pm 0.5\%$ | $76.5\% \pm 0.5\%$ | $79.4\% \pm 0.4\%$ |
| MLP | $94.7\% \pm 0.6\%$ | $58.2\% \pm 1.0\%$ | $95.4\% \pm 0.2\%$ | $76.0\% \pm 0.3\%$ | $76.7\% \pm 0.8\%$ | $80.2\% \pm 0.5\%$ |

Table 10: ADCOL with different number of discriminators.

| Number of discriminators | 1 | 2 | 10 | 20 |
|---|---|---|---|---|
| MNIST | $94.7\% \pm 0.6\%$ | $95.1\% \pm 0.6\%$ | $91.0\% \pm 0.3\%$ | $88.6\% \pm 0.8\%$ |
| SVHN | $58.2\% \pm 1.0\%$ | $45.4\% \pm 1.1\%$ | $52.4\% \pm 1.2\%$ | $46.5\% \pm 1.4\%$ |
| USPS | $95.4\% \pm 0.2\%$ | $95.2\% \pm 0.1\%$ | $95.3\% \pm 0.2\%$ | $90.1\% \pm 0.4\%$ |
| SynthDigit | $76.0\% \pm 0.3\%$ | $73.4\% \pm 0.4\%$ | $67.2\% \pm 0.8\%$ | $73.4\% \pm 0.5\%$ |
| MNIST-M | $76.7\% \pm 0.8\%$ | $76.4\% \pm 0.7\%$ | $72.1\% \pm 0.9\%$ | $57.0\% \pm 1.2\%$ |
| AVG | $80.2\% \pm 0.5\%$ | $77.1\% \pm 0.6\%$ | $75.6\% \pm 0.8\%$ | $71.1\% \pm 1.0\%$ |

ful computation resources than the parties. Thus, the computation overhead in the server side is affordable.

### B.10 SHARING THE PREDICTOR LAYERS

The parties only send the representations to the server in ADCOL. While ADCOL aims to learn a common representation distribution $\mathbf{z}$, an interesting extension is to share the predictor layers between the parties and the server, which ideally helps in regularizing $p(y|\mathbf{z})$. The results are shown in Table 14. We can observe that ADCOL without sharing the predictor layers is generally more effective than sharing the predictor layers. In practice, the distribution $p(y|\mathbf{x}_i)$ is not exactly the same across parties. Thus, it is not necessary to regularize $p(y|\mathbf{z})$ among the parties. Leaving the parties to fine tune their own predictor layer is more capable to learn the personalized local distribution.

### B.11 REPRESENTATION SAMPLING

Here we apply the representation sampling technique and change the sampling rate from $\{20\%, 60\%, 80\%, 100\%\}$. The final accuracy and the communication efficiency are shown in Table 15. We can observe that the communication cost of ADCOL can be significantly reduced with representation sampling. Moreover, there is little accuracy loss when the sampling rate is large than 60%.

### B.12 DIFFERENTIAL PRIVACY

We consider two popular threat models in existing FL studies: 1) The server is trusted and the parties are honest-but-curious (Geyer et al., 2017). We need to protect the messages that are sent from the server to the parties. 2) The server and the parties are honest-but-curious and we need to protect all the transferred messages (Wei et al., 2020; Truex et al., 2020).

**Trusted Server** In this setting, we do not need to protect the representations sent from parties to the server. We need to protect the classification model sent from the server to the parties. Thus, when training the classification model on the server-side, we apply DP-SGD (Abadi et al., 2016)

Table 11: The test accuracy of ADCOL with different representation dimensions.

| Dimension | 512 | 1024 | 2048 |
|---|---|---|---|
| MNIST | $93.2\% \pm 0.6\%$ | $94.9\% \pm 0.5\%$ | $94.7\% \pm 0.6\%$ |
| SVHN | $48.9\% \pm 1.2\%$ | $50.5\% \pm 1.4\%$ | $58.2\% \pm 1.0\%$ |
| USPS | $94.8\% \pm 0.2\%$ | $95.5\% \pm 0.3\%$ | $95.4\% \pm 0.2\%$ |
| SynthDigit | $72.9\% \pm 0.8\%$ | $79.4\% \pm 0.7\%$ | $76.0\% \pm 0.3\%$ |
| MNIST-M | $68.9\% \pm 0.9\%$ | $73.1\% \pm 0.7\%$ | $76.7\% \pm 0.8\%$ |
| AVG | $75.7\% \pm 0.7\%$ | $78.7\% \pm 0.7\%$ | $80.2\% \pm 0.5\%$ |

Table 12: The test accuracy of different approaches on non-IID label settings.

|  | SOLO | FedAvg | FedBN | ADCOL |
|---|---|---|---|---|
| CIFAR-10 | $59.3\% \pm 8.0\%$ | $89.0\% \pm 2.4\%$ | $90.7\% \pm 2.1\%$ | $79.2\% \pm 4.7\%$ |
| CIFAR-100 | $33.5\% \pm 2.1\%$ | $57\% \pm 2.8\%$ | $55.6\% \pm 3.0\%$ | $36.1\% \pm 2.8\%$ |

Table 13: The total training time of running all approaches for 100 rounds.

|  | FedAvg | FedBN | PartialFed | ADCOL | FedProx | Per-FedAvg | FedRep |
|---|---|---|---|---|---|---|---|
| Training time (hour) | 6.5 | 7 | 7 | 11 | 8 | 7 | 8.5 |

to add Gaussian noises to the gradients during training to satisfy $(\varepsilon, \delta)$-DP with the same default parameters. We keep $\varepsilon$ fixed and ensure that $\delta \leq 10^{-2}$ to compare the accuracy of DP-ADCOL with the non-private version as shown in Table 16. We can observe that DP-ADCOL can achieve a very close accuracy to the non-private version with a budget 5. There are two reasons that DP works well in ADCOL: 1) DP-SGD works well with the discriminator as it is a shallow model (Tramer & Boneh, 2021), which is a simple 2-layer MLP. If we increase the number of layers for the discriminators, the accuracy of DP-ADCOLL will decrease as shown in the last row of Table 16. 2) The discriminator needs a small number of steps to update and the accumulated privacy loss is small. For FedAvg, it is not easy to apply record-level DP. Existing studies (Geyer et al., 2017; McMahan et al., 2017) clip the local model updates to provide party-level DP which is more strict than the record-level DP. We conduct simple experiments and find that the accuracy of DP-FedBN is low with party-level DP, which is about 66.3% accuracy given the budget 5.

**Honest-but-curious Server** In this setting, the messages sent from parties to the server should also be protected. We apply local differential privacy with sampling in ADCOL to provide rigorous privacy guarantees. Specifically, in each round, we sample and normalize the representations and add noises from $Gau(0, 1/\epsilon)$ before sending them to the server, where $Lap(0, 1/\epsilon)$ is the Laplace distribution with mean 0 and scale $1/\epsilon$. Then, in each round, the transferred representations satisfy $\epsilon$-differential privacy (Lyu et al., 2020). Due to the parallel composition, the privacy loss is not accumulated among rounds. To achieve the same level of privacy guarantee with DP-ADCOL for FedBN, we implement DP-FedBN by clipping and adding Laplace noises to the communicated model updates (Kairouz et al., 2019). For DP-FedBN, we try two methods: 1) without party sampling: the privacy loss is accumulated among different rounds. 2) party sampling without replacement: we set the sampling fraction per round to 0.2 and the privacy loss is not accumulated among every five rounds. The results are shown in Table 17. We can observe that the accuracy of ADCOL is very close to the non-private version with a modest privacy budget (i.e., 10). Moreover, DP-ADCOL achieves a

Table 14: The comparison between sharing the predictor layers and not sharing the predictor layers.

|  | w/ sharing | w/o sharing |
|---|---|---|
| MNIST | **95.4**% | 94.7% |
| SVHN | 48.8% | **58.2**% |
| USPS | 95.2% | **95.4**% |
| SynthDigit | 75.9% | **76.0**% |
| MNIST_M | 73.4% | **76.7**% |
| AVG | 77.7% | **80.2**% |

Table 15: ADCOL with different representation sampling rates. We present the final converged mean accuracy and the number of communication rounds and communication costs to achieve the target accuracy 78% on Digit.

| Sampling ratio | accuracy | #rounds | size (GB) |
|---|---|---|---|
| 20% | 78.4% | 32 | 0.07 |
| 40% | 78.5% | 32 | 0.14 |
| 60% | 79.1% | 26 | 0.17 |
| 80% | 79.8% | 23 | 0.20 |
| 100% | 80.2% | 21 | 0.23 |

Table 16: The privacy-accuracy tradeoff of DP-ADCOL in the trusted server setting.

| | MNIST | SVHN | USPS | SynthDigit | MNIST-M | AVG |
|---|---|---|---|---|---|---|
| non-private | 94.7% | 58.2% | 95.4% | 76.0% | 76.7% | 80.2% |
| $\varepsilon = 2$ | 90.2% | 52.4% | 90.1% | 67.4% | 69.7% | 74.0% |
| $\varepsilon = 5$ | 93.9% | 57.6% | 94.1% | 72.4% | 73.8% | 78.4% |
| $\varepsilon = 10$ | 94.2% | 57.8% | 94.5% | 74.1% | 74.8% | 79.1% |
| $\varepsilon = 5$ (5 layers) | 91.1% | 53.8% | 91.8% | 69.9% | 72.2% | 75.8% |

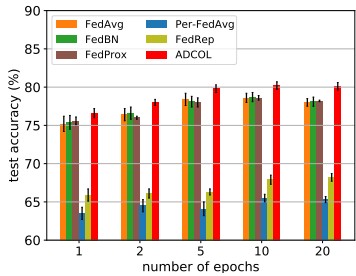

Figure 7: Effect of number of local epochs.

higher accuracy than DP-FedBN in the same privacy level. It is promising to apply DP in ADCOL thanks to the representation-sharing scheme.

Table 17: Comparison between DP-ADCOL and DP-FedBN under the same privacy level.

| $\varepsilon$ | approaches | MNIST | SVHN | USPS | SynthDigit | MNIST_M | AVG |
|---|---|---|---|---|---|---|---|
| 2 | DP-ADCOL | 89.1% | 50.4% | 88.2% | 64.1% | 66.8% | **71.7%** |
| | DP-FedBN (w sampling) | 69.9% | 19.2% | 82.3% | 15.5% | 17.1% | 40.8% |
| | DP-FedBN (w/o sampling) | 76.1% | 21.2% | 78.2% | 10.2% | 10.9% | 39.3% |
| 5 | DP-ADCOL | 92.7% | 56.7% | 93.0% | 69.4% | 71.9% | **76.7%** |
| | DP-FedBN (w sampling) | 73.5% | 23.3% | 85.8% | 28.6% | 26.2% | 47.5% |
| | DP-FedBN (w/o sampling) | 78.7% | 24.4% | 82.5% | 13.2% | 13.7% | 42.5% |
| 10 | DP-ADCOL | 93.1% | 56.9% | 93.1% | 73.2% | 74.3% | **78.1%** |
| | DP-FedBN (w sampling) | 90.5% | 34.3% | 91.2% | 52.8% | 58.7% | 65.5% |
| | DP-FedBN (w/o sampling) | 84.9% | 21.5% | 88.7% | 15.6% | 25.8% | 47.3% |
| non-private | ADCOL | 94.7% | 58.2% | 95.4% | 76.0% | 76.7% | **80.2%** |
| | FedBN | 94.1% | 59.9% | 94.1% | 73.9% | 71.3% | 78.7% |

## B.13 Number of Local Epochs

We vary the number of local epochs $E \in \{1, 2, 5, 10, 20\}$ and report the results in Figure 7. We run all approaches for 100 rounds. If the number of local epochs is too small, the local update is small in each round and the convergence speed is slow. Thus, the accuracy of all approaches is relatively low after running for 100 rounds with a small number of local epochs. ADCOL still consistently outperforms the other approaches with a different number of epochs.

## B.14 Party Sampling with a Fixed Number of Selected Parties

One practical concern is that the output dimension of the discriminator is fixed to be the number of participating parties, which may not handle the case when the number of parties is extremely large or the number of parties is changing over time. To address the concern, we propose to apply party sampling with a fixed number of selected parties each round. The output dimension of the discriminator is same as the number of participated parties each round. The selected parties first update their models locally without the regularization term we introduced. Next, the parties send their representations to the server, which updates the discriminator and sends back the discriminator to the parties. Then, the same parties update their models again with the regularization term using the discriminator. After that, we can move into next round and sample new parties again. We

Table 18: The mean test accuracy and standard derivation across parties when applying party sampling with a fixed number of selected parties each round. The output dimension of the discriminator in ADCOL is set to the number of selected parties each round.

|  | FedAvg | FedBN | ADCOL |
|---|---|---|---|
| Accuracy | 13.4+-3.8% | 13.7+-3.5% | 69.2+-20.6% |

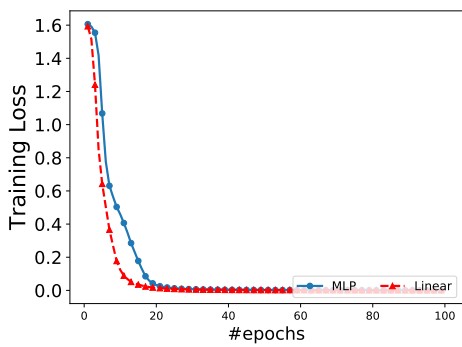

Figure 8: The training curves of different discriminators.

have conducted experiments on Digits with 50 parties. In each round, we randomly drop 5 parties tentatively for one round to simulate the scenario where the number of parties change over time (i.e., the selected 5 parties leave FL for current round and join FL again in the next round). After dropping 5 parties, we randomly select 5 parties to participate in FL in the current round. The output dimension of the discriminator is set to 5. We run ADCOL, FedAvg, and FedBN for 100 rounds and the results are shown in Table 18. We can observe that FedAvg and FedBN have a poor accuracy in such a scenario. ADCOL significantly outperforms these two approaches.

### B.15 CONVERGENCE OF THE DISCRIMINATOR

We empirically study whether the discriminator converge to optima or not. Besides using a MLP as the discriminator in our experiments, to compare convex and non-convex loss function, we also try a linear function as the discriminator by removing the non-linear activation in MLP. The training curves are shown in Figure 8 and the accuracy of using a linear function is shown in Table 19. We can observe both MLP and linear function can achieve optima (i.e., zero training loss) with SGD. Moreover, ADCOL with a linear function as the discriminator can still achieve a better performance than the other baselines from Table 1 of the main paper.

### B.16 STUDY ON THE LOCAL MODEL ARCHITECTURE

Instead of using ResNet-50, we try a different local model to investigate the robustness of our approach. We use the same model as the experiments in FedBN for Digit task, which is a six-layer convolutional neural network. We use the input before the last fully-connected layer as the representation. The results are shown in Table 20. From the table, we can observe that ADCOL outperforms FedBN, which further verifies the effectiveness of ADCOL.

Table 19: ADCOL with MLP or linear function as the discriminator.

| Discriminator | MNIST | SVHN | USPS | SynthDigit | MNIST-M | AVG |
|---|---|---|---|---|---|---|
| Linear Function | 94.9% | 48.4% | 95.7% | 81.9% | 75.9% | 79.3% |

Table 20: Comparison between ADCOL and FedBN using a CNN as local model.

|        | MNIST  | SVHN   | USPS   | SynthDigit | MNIST-M | AVG    |
|--------|--------|--------|--------|------------|---------|--------|
| ADCOL  | **96.6%** | 73.0%  | **97.3%** | **88.3%** | **85.0%** | **88.1%** |
| FedBN  | 96.3%  | **74.8%** | 96.8%  | 85.4%      | 81.8%   | 87.0%  |

Table 21: The comparison between SOLO, ADCOL, and FADA on Digits.

|       | MNIST | SVHN  | USPS  | SynthDigit | MNIST_M | AVG   |
|-------|-------|-------|-------|------------|---------|-------|
| SOLO  | 87.9% | 34.8% | 94.8% | 63.0%      | 67.2%   | 69.5% |
| ADCOL | 94.7% | 58.2% | 95.4% | 76.0%      | 76.7%   | 80.2% |
| FADA  | 85.6% | 40.1% | 89.6% | 68.8%      | 60.5%   | 68.9% |

## C    DISCUSSION

We can consider each party as a domain and studies on multi-domain are also potentially applicable. We have compared ADCOL with the most related FL study on the multi-domain setting (i.e., PartialFed Sun et al. (2021)). Besides PartialFed, we also discuss the relation between ADCOL and the studies on domain adaptation and domain generalization below.

**Relation to Domain Adaptation**    Domain adaptation aims to train a model on a source domain (or multi-source domain), which has a good accuracy on a target domain. A classic and popular approach in domain adaptation is to perform adversarial training, i.e., training a discriminator to encourage domain-invariant features (Ganin et al., 2016; Peng et al., 2019b). Peng et al. (2019b) proposed FADA, which extends domain adaptation in a federated setting. One connection between our approach and domain adaptation is that each party can be viewed as a source domain, and the target domain is the unknown oracle optimal (like domain generalization introduced in Section 3.2 of the main paper). Then, our approach is to extract domain-invariant features from multiple source domain, which is used to regularize the training. To highlight the differences between our approach and the domain adaptation techniques, we compare our approach with the federated domain adaptation study (Peng et al., 2019b) (FADA) and show the main differences: (1) Setting: FADA aims to train a model on multiple source domain, which has a good accuracy on a target domain. Our study aims to train a personalized model for each party, which has good accuracy on its local data. (2) Discriminator: FADA uses multiple discriminators, where each discriminator is used for binary classification for one source-target domain pair. Our study uses a single discriminator for the multi-classification among all parties. Moreover, we have provided the theoretical analysis on the convergence properties. (3) Framework: FADA uses adversarial training to generate domain-invariant and domain-specific features. Our study uses adversarial training to regularize the local training in federated learning.

Intuitively, we cannot directly compare ADCOL and FADA in the experiments since the settings are different. In our experiments, there is no a single target domain for testing in FADA. One method is to treat each party as a target domain and applying FADA $N$ times, where $N$ is the number of parties. However, the computation and communication overhead is significantly large. Moreover, such an approach does not utilize the labels of the target dataset. We have compared ADCOL and FADA using the above method and the results are shown in Table 21. ADCOL significantly outperforms FADA. Moreover, the test accuracy of FADA is even smaller than local training in many cases since it does not exploit the labels of the target dataset.

**Relation to Domain Generalization**    While the motivation of ADCOL is intuitive, it can also be explained from the perspective of domain generalization (Muandet et al., 2013). In domain generalization, the goal is to extract knowledge from multiple source domains to apply it to an unseen target domain. Considering each party as a source domain and the target domain as the oracle optimal representation space, we aim to extract the domain invariant representation distribution and use it to regularize the local training. Existing domain generalization techniques are designed in a centralized setting, which usually require the access to the raw data of multiple source domains (Li et al., 2018a;b; Liu et al., 2018). There is one work (Liu et al., 2021) that studies domain generalization in the federated setting. It is designed for medical image segmentation by episodic learning in the frequency

space. In this paper, we aim to design a general collaborative learning framework based on adversarial learning.

**Limitations**   ADCOL is a collaborative learning method for non-IID features. As shown in Appendix B.8, the performance of ADCOL is poor compared with federated learning approaches on non-IID label setting. Note that ADCOL aims to learn a common representation distribution. Intuitively, the task-specific representations of images from different classes should be very different, which can be easily classified by a small MLP. Thus, if the label distribution varies across parties, the representation distribution naturally also varies a lot across parties. The current objective of ADCOL does not fit into the non-IID label setting.

As shown in Section 3.5 of the main paper, the communication size of ADCOL is related to the number of examples. If the number of examples is very large and the size of the model is small, the communication cost of ADCOL will be larger than other federated learning approaches. However, local training can usually achieve satisfactory performance if the dataset size is very large. In such cases, besides ADCOL, existing federated learning approaches may also not help.

**Insights and Future Work**   The key insights from ADCOL are (1) a GAN-style training scheme and (2) regularization from a view of representation distribution. While ADCOL does not have a requirement on the vanilla local training algorithm, it can also be extended to self-supervised federated learning, where the cross-entropy loss is replaced by the loss used in self-supervised learning (e.g., contrastive loss (Chen et al., 2020; Chen & He, 2021)). Moreover, while ADCOL only works on non-IID feature settings currently, the adversarial collaborative training scheme can potentially be applied to address other data settings by modifying the objectives of local training and server training. There are many research opportunities based on the findings of this paper.

