# OpenReview forum: "Adversarial Collaborative Learning on Non-IID Features"
_ICLR.cc/2023/Conference — Submitted to ICLR 2023_

### Official Review · Reviewer_bkH5 · 2022-10-21

**Confidence:** 5
**Correctness:** 3
**Technical Novelty And Significance:** 3
**Empirical Novelty And Significance:** 2
**Recommendation:** 6

**Clarity, Quality, Novelty And Reproducibility:**

In general, it is well-organized and written, and the experimental results are encouraging. I'm not sure if Reproducibility is possible


**Strength And Weaknesses:**

Strength:
In this paper, focusing on the Non-IID feature, the author thinks out of the model-averaging scheme used in FL, and proposes a novel learning concept called adversarial collaborative learning. Instead of averaging the local models, the author applies adversarial learning to match the representation distributions of different parties. Specifically, the server aims to train a discriminator to distinguish the local representations by the party IDs, while the parties train the base encoders such that the generated representations cannot be distinguished by the discriminator. Besides the base encoders, each party trains a predictor for local personalization and ensures that the generated representation is meaningful for the prediction task.

Weakness:
The authors claim they used the same task as in the FedBN study. However, the experimental setup in the paper does not exactly FOLLOW the setup of FEDBN, which may give the impression that the authors are choosing the experimental setup that is more favorable to their own method. More importantly, FedBN was designed for the Non-IID Feature task, but the experimental results in this paper yielded a performance inconsistent with FedBN. In addition, while this paper proposed a novel collaborative learning approach for non-IID features based on adversarial architecture and claimed to achieve better performance than other SOTA methods. However, it is known that adversarial architectures are challenging to train, and thus may be weak in experimental deployment and replication compared to other FL methods for Non-IID features.


**Summary Of The Paper:**

The paper proposed a new learning concept called ADCOL(Adversarial Collaborative Learning) for non-IID features. Instead of adopting the widely used model-averaging scheme, ADCOL conducts training in an adversarial way: the server aims to train a discriminator to distinguish the representations of the parties, while the parties aim to generate a common representation distribution.


**Summary Of The Review:**

In this paper, focusing on the Non-IID feature, the author thinks out of the model-averaging scheme used in FL, and proposes a novel learning concept called adversarial collaborative learning.
1. The authors should explain why they did not follow the FedBN experiments completely. Of course, it would be better if the authors could give the performance of the complete follow FedBN experimental setup to demonstrate the advantages of the proposed method.
2. In the experimental results analysis section, the authors should give more analysis of the methods designed for the Non-IID feature, especially FedBN. For example in Table 16, the effectiveness of the proposed method on Non-IID features might be better demonstrated if the authors could give the results of DP-FedBN.

---

> ### Author Response · Authors · 2022-11-15
> **Response to Reviewer bkH5**
>
> Thanks a lot for your comments!
>
> > Q1. The experimental setup in the paper does not exactly FOLLOW the setup of FEDBN. The authors should explain why they did not follow the FedBN experiments completely. Of course, it would be better if the authors could give the performance of the complete follow FedBN experimental setup to demonstrate the advantages of the proposed method.
>
> There are two major differences between our experimental setup and the setup in FedBN.
> * The model architecture is different. Our paper adopts ResNet-50 for all datasets, while FedBN uses a simple CNN for Digit and AlexNet for Office and DomainNet. We adopt ResNet since we need to extract the representations for the input data, and ResNet-50 is commonly used as a base encoder to extract the representations in representation learning studies [1,2,3].
> [1] A Simple Framework for Contrastive Learning of Visual Representations ICML 2020
> [2] Bootstrap your own latent: A new approach to self-supervised Learning Neurips 2020
> [3] Exploring Simple Siamese Representation Learning CVPR 2021
> * The image size is different. FedBN resizes all images of Office and Domain to 256x256x3, while our paper resizes them to 64x64x3 for computation efficiency.
>
> We have added the explanation to **Appendix B.1** of the revised paper.
>
> We also add experiments to use the same experimental setup as FedBN on Digit task. The results are shown in **Table 20** of the revised paper (also shown below). ADCOL still outperforms FedBN by 1.1% on average in such a setting, which further verifies the effectiveness of ADCOL.
>
> |       | MNIST | SVHN | USPS | SynthDigit | MNIST-M | AVG  |
> |-------|-------|------|------|------------|---------|------|
> | ADCOL | 96.6  | 73.0   | 97.3 | 88.3       | 85.0      | 88.1 |
> | FedBN | 96.3  | 74.8 | 96.8 | 85.4       | 81.8    | 87.0 |
>
> > Q2. Adversarial architectures are challenging to train, and thus may be weak in experimental deployment and replication compared to other FL methods for Non-IID features.
>
> We have provided all the experimental details for reproducibility. Moreover, we will provide the code if the paper is accepted.
>
>
> > Q3. In the experimental results analysis section, the authors should give more analysis of the methods designed for the Non-IID feature, especially FedBN. For example in Table 16, the effectiveness of the proposed method on Non-IID features might be better demonstrated if the authors could give the results of DP-FedBN.
>
> We have added more analysis of FedBN in the revised paper. Specifically, we have added the results of DP-FedBN in **Table 17** of the revised paper (also shown below). DP-ADCOL outperforms DP-FedBN. We have also added the results of FedBN of non-IID label setting in **Table 12**. FedBN works like FedAvg in the non-IID label setting since it averages most of the model parameters. Now FedBN is included in all experiments that compare different FL approaches.
>
> | $\varepsilon$ | approaches               | MNIST | SVHN  | USPS  | SynthDigit | MNIST_M | AVG   |
> |--------------|--------------------------|-------|-------|-------|------------|---------|-------|
> | 2            | DP-ADCOL                 | 89.1% | 50.4% | 88.2% | 64.1%      | 66.8%   | **71.7%** |
> |              | DP-FedBN (w sampling)   | 69.9% | 19.2% | 82.3% | 15.5%      | 17.1%   | 40.8% |
> |              | DP-FedBN (w/o sampling) | 76.1% | 21.2% | 78.2% | 10.2%      | 10.9%   | 39.3% |
> | 5            | DP-ADCOL                 | 92.7% | 56.7% | 93.0% | 69.4%      | 71.9%   | **76.7%** |
> |              | DP-FedBN (w sampling)   | 73.5% | 23.3% | 85.8% | 28.6%      | 26.2%   | 47.5% |
> |              | DP-FedBN (w/o sampling) | 78.7% | 24.4% | 82.5% | 13.2%      | 13.7%   | 42.5% |
> | 10           | DP-ADCOL                 | 93.1% | 56.9% | 93.1% | 73.2%      | 74.3%   | **78.1%** |
> |              | DP-FedBN (w sampling)   | 90.5% | 34.3% | 91.2% | 52.8%      | 58.7%   | 65.5% |
> |              | DP-FedBN (w/o sampling) | 84.9% | 21.5% | 88.7% | 15.6%      | 25.8%   | 47.3% |
> | non-private  | ADCOL                    | 94.7% | 58.2% | 95.4% | 76.0%      | 76.7%   | 80.2% |
> |              | FedBN                   | 94.1% | 59.9% | 94.1% | 73.9%      | 71.3%   | 78.7% |

---

> > ### Comment · Reviewer_bkH5 · 2022-11-23
> > **My concerns have been addressed**
> >
> > My concerns have been addressed and recommended for acceptance

---

> > > ### Author Response · Authors · 2022-11-24
> > > **Thanks for your response!**
> > >
> > > We are happy that our revision addresses your concerns. Thanks a lot for your response and support!

---

> ### Author Response · Authors · 2022-11-17
> **Author-Reviewer Discussion Due Approaching**
>
> Dear Reviewer bkH5,
>
> Thanks for your comments! The Author-Reviewer discussion stage will close in about two days. We have addressed all your comments, including the experimental setup concern, replication issue, and analyses on FedBN. We would appreciate it if you could read our response, provide further feedback, and reconsider your ratings if appropriate. Thanks a lot!

---

### Official Review · Reviewer_h9oS · 2022-10-24

**Confidence:** 4
**Correctness:** 3
**Technical Novelty And Significance:** 2
**Empirical Novelty And Significance:** 3
**Recommendation:** 5

**Clarity, Quality, Novelty And Reproducibility:**

The paper presentation is clear, and the organization is reasonable. The novelty is incremental, and the reproducibility seems ok from shown context details.

**Strength And Weaknesses:**

[Strength]
The problem of the paper aims to address is practical, non-iid distribution is a key factor in the real-world federated learning setting. The motivation is reasonable.
The structure of the paper is complete and well-organized.
The experiments are extensive.

[Weakness]
The theoretical contribution of the paper is incremental. The proposed ADCOL learning framework is not novel to some extent. The similar structure also exists in many GAN-based multi-task and multi-view learning settings, i.e., the generator aims to capture the domain-invariant (task-invariant or view-invariant) knowledge, and the discriminator aims to detect which encoding comes from which domain (task, or view). Bringing this similar idea to the federated learning setting may limit the contribution of the paper.

The awareness of transferring the representation other than model parameters is good in terms of privacy leaks. But the solution is straightforward. It could be more helpful to do the in-depth analysis to enhance the paper more solid.

For the selected datasets, it is better to show the statistical distribution of the heterogeneous features. It could be more aligning the main idea of the paper and help to explain the following experimental results. For the current version, the discussion of the outperformance is limited. Just listing the numbers may not be sufficient.

In Table 2, the communication round and communication size for Per-FedAVG and FedRep are missing. The authors may want to explain why they are not reported.

**Summary Of The Paper:**

In this paper, the authors aim to address the non-IID feature distribution in the federated learning setting. To achieve this goal, the authors propose Adversarial Collaborative Learning (ADCOL) by introducing a discriminator in the center server and encoders among parties. For the proposed ADCOL, the authors do the theoretical analysis and extensive experiments to show the effectiveness. Overall, the structure paper is well-organized, and the contribution is incremental.

**Summary Of The Review:**

Overall, the paper introduces a solution for addressing the non-IID feature distribution problem in the federated learning setting. The paper is complete with the proposed method, theoretical analysis, and experiments. The novelty is incremental, and some details can be enhanced.

---

> ### Author Response · Authors · 2022-11-15
> **Response to Reviewer h9oS (1/2)**
>
> Thanks a lot for your comments!
>
> > Q1. The theoretical contribution of the paper is incremental. The proposed ADCOL learning framework is not novel to some extent. The similar structure also exists in many GAN-based multi-task and multi-view learning settings, i.e., the generator aims to capture the domain-invariant (task-invariant or view-invariant) knowledge, and the discriminator aims to detect which encoding comes from which domain (task, or view). Bringing this similar idea to the federated learning setting may limit the contribution of the paper.
>
> To the best of our knowledge, we are the first to propose a new collaborative learning framework based on adversarial learning instead of model averaging. We agree with you that adversarial learning itself is not new. However, the novelty of the paper is to have a novel application of adversarial learning to collaborative learning, which is non-trivial. **Our paper shows that it is promising to use a simple and neat framework based on adversarial learning to conduct FL instead of model-averaging, which is a fundamentally new way in FL**. Thus, this gives a new rethinking about how to design effective collaborative learning beyond the mainstream model averaging methods. We believe our paper can bring new insights to some readers, e.g., Reviewer bkH5 thinks our approach is novel.
>
> Specifically, applying adversarial learning to capture domain-invariant knowledge has been applied in other areas such as domain adaptation and domain generalization. However, these studies usually need to access all the raw data and it is non-trivial to apply them in the federated setting in a neat way. For example, FADA [1] extends domain adaptation in the federated setting using adversarial learning. It trains multiple discriminators, where each discriminator is used for binary classification for one source-target domain pair, which is complicated and costly. For more discussion on the comparison between our approach and domain adaptation/generalization, please refer to **Appendix C**. We’re also grateful if you could provide any related reference that we may add for discussion.
>
> [1] Federated adversarial domain adaptation.
>
> > Q2. The awareness of transferring the representation other than model parameters is good in terms of privacy leaks. But the solution is straightforward. It could be more helpful to do the in-depth analysis to enhance the paper more solid.
>
> We did not focus on privacy in our paper so we provided a simple and effective solution to apply DP to protect the representations. We have added more in-depth analyses in **Appendix B.12** of the revised paper. Specifically, we consider two different threat models: trusted server and honest-but-curious server. In the first case, we apply DP-SGD [2] for the training of the discriminator. The results are shown in **Table 16** of the revised paper (also shown below). DP-SGD works well in our approaches thanks to **1) the shallow model used as the discriminator in our approach and 2) the small number of steps required to update the discriminator**. In the second case, we apply LDP with sampling to protect the transferred representations. The results are shown in **Table 17** of the revised paper. We can observe that DP-ADCOL outperforms DP-FedBN. Since we transfer representations instead of models, each party can sample a fraction of representations and send them to the server. As long as there is no overlap between the sampled data of two rounds, the privacy loss is not accumulated due to parallel composition. In summary, **with additional analyses and experiments, we find that our training framework can work well with DP due to the representation sharing and shallow model training in the server**.
>
> |                       | MNIST  | SVHN   | USPS   | SynthDigit | MNIST-M | AVG             |
> |-----------------------|--------|--------|--------|------------|---------|-----------------|
> | non-private           | 94.7% | 58.2% | 95.4% | 76.0%     | 76.7%  | 80.2% |
> | $\varepsilon=2$            | 90.2% | 52.4% | 90.1% | 67.4%     | 69.7%  | 74.0%          |
> | $\varepsilon=5$             | 93.9% | 57.6% | 94.1% | 72.4%     | 73.8%  | 78.4%          |
> | $\varepsilon=10$            | 94.2% | 57.8% | 94.5% | 74.1%     | 74.8%  | 79.1%          |
> | $\varepsilon=5$  (5 layers) |   91.1%     |   53.8%     | 91.8%       | 69.9%           |72.2%         |   75.8%              |
>
> For the comparison between the information leakage of representations and models, we believe it is a very interesting direction and needs another paper to study. We consider studying it comprehensively in future work.

---

> > ### Author Response · Authors · 2022-11-15
> > **Response to Reviewer h9oS (2/2)**
> >
> > > Q3. For the selected datasets, it is better to show the statistical distribution of the heterogeneous features. It could be more aligning the main idea of the paper and help to explain the following experimental results.
> >
> > **We indeed provided the distribution of the features to explain the experimental results in Appendix B.1 and B.2 of the initial version**. We browse the literature and we find that FID [3] is a good metric to measure the difference between data distributions. Specifically, it measures the Frechet distance between the representation distributions of different datasets, where the representation is generated by an Inception v3 model pretrained on ImageNet dataset. FID is 0 when two datasets are the same. We present FID in Table 1 of Appendix B.1, which shows that the feature distributions of different datasets are quite different. Specifically, the FID for Digits, Caltech-10, and DomainNet are 140.97, 78.12, and 144.81, respectively. Besides FID, we also use t-SNE to visualize the feature distribution. We train a ResNet-50 on all datasets (i.e., parties) from a task and extract the feature distributions of each dataset. Then, we use t-SNE to visualize the representation as shown in **Figure 4**. We can clearly observe that the feature distribution of each party is different.
> >
> > [3] Gans trained by a two time-scale update rule converge to a local nash equilibrium. NIPS 2017.
> >
> > We have made a very interesting observation **there is a correlation between FID and the performance gain of ADCOL compared with local training**. Generally, with a higher FID (i.e., more imbalanced feature distributions), the party can gain more from our approach. The relative improvement in the accuracy of ADCOL against local training is 15.4%, 6.8%, and 17.2% on Digits, Caltech-10, and DomainNet, respectively. The improvement is positively related to the FID of each task. If FID is small, the representation distribution of the local dataset is close to the global dataset, then local training may already learn a good representation and the improvement of ADCOL is limited. We have highlighted it in **Appendix B.3** of the revised paper.
> >
> >
> > > Q4. In Table 2, the communication round and communication size for Per-FedAVG and FedRep are missing. The authors may want to explain why they are not reported.
> >
> > We explain the reason in the caption of Table 2. Table 2 presents the communication round and size of each approach to achieve the same target performance. Since the performances of Per-FedAvg and FedRep are poor as shown in Table 1, they cannot achieve the target performance and we use slash cells to indicate such cases.

---

> > > ### Comment · Reviewer_h9oS · 2022-12-04
> > > **Thank you for the response!**
> > >
> > > I appreciate the authors' detailed response to my comments. However, my major concern in terms of the limited novelty and lack of in-depth analysis remains. In particular, none of the 3 main theorems is directly addressing the non-IID features. Therefore, I would like to keep my original scores.

---

> > > > ### Author Response · Authors · 2022-12-05
> > > > **Further response**
> > > >
> > > > Dear Reviewer h9oS,
> > > >
> > > > Thanks for your reply.
> > > >
> > > > > Q1. The limited novelty.
> > > >
> > > > We have highlighted that our paper is the first one to use a simple and neat framework based on adversarial learning to conduct FL instead of model-averaging, which is a fundamentally new way in FL. We think whether a paper is novel or not is quite subjective and a paper is valuable as long as it brings new insights to the community.
> > > >
> > > >
> > > > > Q2. Lack of in-depth analysis. In particular, none of the 3 main theorems is directly addressing the non-IID features.
> > > >
> > > > In your original review, you said “The awareness of transferring the representation other than model parameters is good in terms of privacy leaks. But the solution is straightforward. It could be more helpful to do the in-depth analysis to enhance the paper more solid.” Thus, we added more analysis in the privacy part as shown in Appendix B.12.
> > > >
> > > > The three main theorems do not have any limitations on the data distributions. It is not necessary to develop the theorems only for non-IID features. The theorems analyze the convergence properties of the proposed framework, i.e., the parties will achieve the same representation distribution with the framework, which aligns with our motivation of the proposed design for non-IID features as shown in Section 3.2. **The theorems prove that the framework can achieve our goal (i.e., learning a common representation distribution), and our goal is only applicable to the non-IID feature setting.** If the data is non-IID labels, since the task-specific representations of images from different classes should be very different, the representation distribution naturally varies a lot across parties. Thus, our goal and design are not applicable to non-IID labels as shown in Appendix B.8. Moreover, we have provided analyses on the influence of the feature skew on the benefit of our algorithm in Appendix B.3.
> > > >
> > > > Please let us know what kind of in-depth analysis you want so that we can add it if possible.

---

> ### Author Response · Authors · 2022-11-17
> **Author-Reviewer Discussion Due Approaching**
>
> Dear Reviewer h9oS,
>
> Thanks for your comments! The Author-Reviewer discussion stage will close in about two days. We have addressed all your comments, including the contribution concern, in-depth privacy analysis, and statistics of the datasets to explain the experimental results. We would appreciate it if you could read our response, provide further feedback, and reconsider your ratings if appropriate. Thanks a lot!

---

> ### Author Response · Authors · 2022-11-29
> **Looking forward to your feedback**
>
> Dear Reviewer h9oS,
>
> Thanks again for your review. We'd like to know whether our response addresses your concerns. We are also happy to answer if you have any further questions. Thank you!

---

### Official Review · Reviewer_vpMT · 2022-11-02

**Confidence:** 3
**Correctness:** 3
**Technical Novelty And Significance:** 2
**Empirical Novelty And Significance:** 3
**Recommendation:** 6

**Clarity, Quality, Novelty And Reproducibility:**

Clarity.
This paper is well-written and easy to follow.  This paper explaind the ADCOL clearly and show its better performance in experiments. But the theorems are not clear enough.

Quality. & Novelty.
The Adversarial Collaborative Learning(ADCOL) has been used in other areas and shown good performance. This paper introduces the idea in FL. The method is interesting and new. Also, the paper gives some theoretical analysis and competitive experiemental results. However, averaging results of different datasets seems meaningless.

Reproducibility.
This paper does not provide code and data for reproducing the results.

**Strength And Weaknesses:**

Strength.
1. This paper is well-written and easy to follow.
2. The method is new in the area of FL.

Weakness.
1. The novelty is limited.
2. The experimental result is not such rigorous.

**Summary Of The Paper:**

This paper proposes a new learning method, ADCOL, for non-iid features for federated learning. Furthermore, the experiments on three tasks show that ADCOL achieves better performance than state-of-the-art FL algorithms on non-IID features.

**Summary Of The Review:**

This paper proposes a new learning method, ADCOL, for non-iid features for federated learning. The Adversarial Collaborative Learning(ADCOL) has been used in other areas and shown good performance. This paper first introduces the idea in FL to solve the non-iid feature problem. Furthermore, the experiments on three tasks show that ADCOL achieves better performance than state-of-the-art FL algorithms on non-IID features.

---

> ### Author Response · Authors · 2022-11-15
> **Response to Reviewer vpMT**
>
>
> Thanks a lot for your comments!
>
> > Q1. The novelty is limited. The Adversarial Collaborative Learning (ADCOL) has been used in other areas and shown good performance. This paper introduces the idea in FL.
>
> To the best of our knowledge, we are the first to propose a new collaborative learning framework based on adversarial learning instead of model averaging. Adversarial learning (not adversarial collaborative learning) has been used in many areas (e.g., data synthesis [1], domain adaptation [2]). However, introducing it to the federated setting as a general framework is non-trivial as the studies in the centralized setting usually need to access all the raw data which is not allowed in the federated setting. There are some studies that also apply adversarial learning in the federated setting. For example, FADA [3] extends domain adaptation in the federated setting using adversarial learning. It trains multiple discriminators, where each discriminator is used for binary classification for one source-target domain pair, which is complicated and costly. Instead, we design a simple and neat framework using adversarial learning without model-averaging. **Our paper gives a new rethinking about how to design effective collaborative learning beyond the mainstream model averaging methods**. We believe our paper can bring new insights to some readers, e.g., Reviewer bkH5 thinks our approach is novel. For more discussion on the comparison between our approach and the other FL studies that apply adversarial learning, please refer to Appendix C.
>
> [1] Generative adversarial networks.
>
> [2] Domain-adversarial training of neural networks.
>
> [3] Federated adversarial domain adaptation.
>
> > Q2. The experimental result is not such rigorous. Averaging results of different datasets seems meaningless.
>
> Note that in our non-IID feature setting, each party has a different dataset of the same task (see the third paragraph of **Section 5.1**). Thus, averaging results of different datasets denote the mean accuracy across parties, which reflects the general effectiveness of the algorithms. Kindly let us know if you still think the averaged result is meaningless and we will remove it in the revision.
>
> > Q3. The theorems are not clear enough.
>
> We have added more explanation about the theorems before Theorem 4.1 in **Section 4.1** of the revised paper. Specifically, in Theorem 4.1, we derive the optimal discriminator given the objective Equation 4. Then, in Theorem 4.2, we derive the optimal solution for the distributions of local representations to minimize the local objective Equation 3 given the optimal discriminator from Theorem 4.1. Last, in Theorem 4.3, we show that the distribution of local representations can converge to the optimal solution given in Theorem 4.2.
>
> > Q4. This paper does not provide code and data for reproducing the results.
>
> We use public datasets for our experiments. Moreover, we will make our code publicly available if the paper is accepted.

---

> ### Author Response · Authors · 2022-11-17
> **Author-Reviewer Discussion Due Approaching**
>
> Dear Reviewer vpMT,
>
> Thanks for your comments! The Author-Reviewer discussion stage will close in about two days. We have addressed all your comments, including the clarification about the theorems, the (possible) misunderstanding about our experimental setup, and the novelty and reproducibility concern. We would appreciate it if you could read our response, provide further feedback, and reconsider your ratings if appropriate. Thanks a lot!

---

> ### Author Response · Authors · 2022-11-29
> **Looking forward to your feedback**
>
> Dear Reviewer vpMT,
>
> Thanks again for your review. We'd like to know whether our response addresses your concerns. We are also happy to answer if you have any further questions. Thank you!

---

### Author Response · Authors · 2022-11-15
**Revision Summary**

We thank all reviewers for their efforts and time in reviewing our paper. We are glad that Reviewer $\textcolor{red}{bkH5}$ found our approach novel, Reviewer $\textcolor{blue}{vpMT}$, $\textcolor{orange}{h9oS}$, and $\textcolor{red}{bkH5}$ found our paper well-written and organized, Reviewer $\textcolor{orange}{h9oS}$ found our experiments extensive, and Reviewr $\textcolor{red}{bkH5}$ found our experimental results encouraging. We have completely addressed your comments, which further improves the paper. Thanks!

The major updates of our revision paper are highlighted in red, which are summarized below.
* **[Section 4.1]** We have added the clarification for the theorems to address Reviewer $\textcolor{blue}{vpMT}$’s comments.
* **[Table 2]** We have highlighted the reason for the missing cells to address Reviewer $\textcolor{orange}{h9oS}$’s comments.
* **[Appendix B.1]** We have highlighted the statistical distributions of the datasets to address Reviewer $\textcolor{orange}{h9oS}$’s comments. We have also added the difference between our experimental setup and the setting in FedBN to address Reviewer $\textcolor{red}{bkH5}$’s comments.
* **[Appendix B.3]** We have added the section to explain our experimental results by the FID score to address Reviewer $\textcolor{orange}{h9oS}$’s comments.
* **[Appendix B.12 & Table 16]** We have added more in-depth experiments and analyses on applying DP to our approach to address Reviewer $\textcolor{orange}{h9oS}$’s comments.
* **[Table 17]** We have added the results of DP-FedBN to address Reviewer $\textcolor{red}{bkH5}$’s comments.
* **[Appendix B.16]** We have added the results of ADCOL and FedBN with the same setting as FedBN to address Reviewer $\textcolor{red}{bkH5}$’s comments.

---

### Author Response · Authors · 2022-11-21
**Discussion**

Dear Reviewers,

We have addressed your concerns and revised the paper accordingly. We have put a lot of effort into the rebuttal and revision. We really look forward to your feedback. Please let us know if you have any further concerns and reconsider your ratings if appropriate. Thanks!

---

> ### Author Response · Authors · 2022-11-23
> **A gentle reminder**
>
> Dear Reviewers,
>
> We believe the discussion is important given the borderline ratings (6, 6, 5). We are looking forward to your feedback. Thanks a lot!

---

### Decision · Program_Chairs · 2023-01-20

**Decision:**

Reject

**Justification For Why Not Higher Score:**

N/A

**Justification For Why Not Lower Score:**

N/A

**Metareview: Summary, Strengths And Weaknesses:**

The paper proposes a new adversarial collaborative learning for non-IID features. The idea is nice and the problem studied in this paper seems practical. However, the technical contribution of the paper is very limited. The proposed framework already exists in the related multi-task and multi-view learning settings. Moreover, the experimental setup seems mismatched FEDBN. The authors are encouraged to improve the paper according to the reviewers comments.